# Inference of B cell clonal families using heavy/light chain pairing information

**Duncan K. Ralph**[1]*, **Frederick A. Matsen IV**[1,2,3,4]

**1** Computational Biology Program, Fred Hutchinson Cancer Research Center, Seattle, Washington, United States of America, **2** Department of Genome Sciences, University of Washington, Seattle, Washington, United States of America, **3** Department of Statistics, University of Washington, Seattle, Washington, United States of America, **4** Howard Hughes Medical Institute, Computational Biology Program, Fred Hutchinson Cancer Research Center, Seattle, Washington, United States of America

* dralph@fredhutch.org

**Data Availability Statement:** Data is available at https://doi.org/10.5281/zenodo.5860143 Code is available at https://github.com/psathyrella/partis/.

**Funding:** This work supported by NIH grants R01 AI146028 (PI Matsen), U01 AI150747 (PI Rustom

## Abstract

Next generation sequencing of B cell receptor (BCR) repertoires has become a ubiquitous tool for understanding the antibody-mediated immune response: it is now common to have large volumes of sequence data coding for both the heavy and light chain subunits of the BCR. However, until the recent development of high throughput methods of preserving heavy/light chain pairing information, these samples contained no explicit information on which heavy chain sequence pairs with which light chain sequence. One of the first steps in analyzing such BCR repertoire samples is grouping sequences into clonally related families, where each stems from a single rearrangement event. Many methods of accomplishing this have been developed, however, none so far has taken full advantage of the newly-available pairing information. This information can dramatically improve clustering performance, especially for the light chain. The light chain has traditionally been challenging for clonal family inference because of its low diversity and consequent abundance of non-clonal families with indistinguishable naive rearrangements. Here we present a method of incorporating this pairing information into the clustering process in order to arrive at a more accurate partition of the data into clonally related families. We also demonstrate two methods of fixing imperfect pairing information, which may allow for simplified sample preparation and increased sequencing depth. Finally, we describe several other improvements to the *partis* software package.

## Author summary

Antibodies form part of the adaptive immune response, and are critical to immunity acquired by both vaccination and infection. Next generation sequencing of the B cell receptor (BCR) repertoire provides a broad and highly informative view of the DNA sequences from which antibodies arise. Until recently, however, this sequencing data was not able to pair together the two domains (from separate chromosomes) that make up a functional antibody. In this paper we present several methods to improve analysis of the new *paired* data that does pair together sequence data for complete antibodies. We first

Antia), as well as R01 AI120961 and R01 AI138709 (PI Julie Overbaugh). The research of FAM was supported in part by a Faculty Scholar grant from the Howard Hughes Medical Institute and the Simons Foundation, and he is an investigator of the Howard Hughes Medical Institute. Scientific Computing Infrastructure at Fred Hutch funded by ORIP grant S10OD028685. The funders had no role in study design, data collection and analysis, decision to publish, or preparation of the manuscript.

**Competing interests:** The authors have declared that no competing interests exist.

show a method that better groups together sequences stemming from the same ancestral cell, solving a problem called "clonal family inference." We then show two methods that can correct for various imperfections in the data's identification of which sequences pair together to form complete antibodies, which together may allow for significantly simplified experimental methods.

This is a *PLOS Computational Biology* Methods paper.

## Introduction

Antibodies are a primary component of the adaptive immune system, and are critical to the operation of immune memory. They are produced by B cells in a process that generates a vast diversity designed to prepare for encounter with any potential antigen. This process begins with somatic rearrangement of germline genes in naive B cells. Perhaps 50–80% of these naive cells are initially self-reactive, but during the course of maturation this is reduced to around 10%, largely due to central tolerance [1]. For B cells that go on to meet their cognate antigen, the process continues in germinal centers as affinity maturation proceeds via somatic hypermutation (SHM) and antigen driven selection.

We can think of this as two steps: first, the generation of an initial repertoire that tries to span the space of potential antigens such that there will be a naive antibody with some level of binding against any antigen. Second, once an antigen is encountered, any B cells producing "nearby" antibodies with some binding evolve "toward" the antigen, resulting in antibodies with higher affinity S1 Fig.

Antibodies consist of two analogous parts, called the heavy and light chains [2] (which we sometimes refer to simply as *heavy* or *light*, omitting the "chain"). To construct a heavy chain, one V, one D, and one J gene are selected randomly (but very non-uniformly) from among the many possibilities on each chromosome. The selected V, D, and J genes are joined together in such a way that at each junction some number of random "non-templated" nucleotides can be added, and a random number of nucleotides can be deleted from each gene's end. Light chain rearrangement is similar, except that it has no D genes, and only limited amounts of insertion and deletion [3]. Light chains are thus vastly less diverse than heavy: perhaps by a factor of $10^6$[4] or more [3]. One rare but potentially significant exception to this basic picture is the use of two D genes (VDDJ rearrangement) [2, 5].

During development, each B cell first attempts a heavy chain rearrangement. If the result has V and J in frame with respect to each other and is free of stop codons, and if it can successfully pair with a special temporary *surrogate* light chain, the cell then goes on to attempt light chain rearrangement. While heavy chain rearrangement appears optimized to maximize diversity, light chain rearrangement instead maximizes the opportunities for generating a functional and non-self-reactive partner for that heavy chain [3, 4]. Light chain genes within a locus, for instance, are found along both strands, so that intervening genes are not always excised in the first rearrangement, and are thus available for subsequent rearrangements. Rearrangement also initially favors the innermost genes in a light locus, such that outer genes are preserved for subsequent attempts. Most species also have two independent light chains (kappa and lambda, commonly called IgK and IgL) on separate chromosomes, which again increases the number

of opportunities to find a successful light chain partner once the resources have been expended to obtain a functional heavy chain (commonly called IgH). These methods of salvaging a potentially failed antibody, referred to as *receptor editing*, while more prevalent in light chain, also occur in heavy, for instance via VH replacement [4, 6].

In both chains, successful rearrangement on one chromosome typically results in suppression of rearrangement on the other; this is called allelic exclusion. The process is, however, sometimes incomplete, resulting in *allelic inclusion*, or expression of multiple heavy or light chains at some frequency, which has been estimated at around 1% of cells [4, 7]. There also exist other less common processes that can result in expression of, for example, up to four light chains in a single cell [8].

The full paired antibody structure is arranged such that both heavy and light chains contribute to, and are necessary for, both binding and structural stability. Each chain has three loops, called *complementarity determining regions* (CDRs), that provide much of the antigen contact and, correspondingly, also most of the diversity. Of the three, two are encoded by the V germline (CDR1 and CDR2), while the third (CDR3) contains all of the junctional diversity generated by VDJ rearrangement. Because SHM is targeted to the CDRs, they also harbor most of the diversity generated during affinity maturation. The intervening framework (FWK) regions, on the other hand, are more important for structural stability, and are thus much less variable.

## Analyzing antibody repertoires

Our ability to measure the B cell receptor (BCR) repertoires that result from the processes described above has improved over time [9]. Beginning in the 1990s, Sanger sequencing allowed the observation of hundreds of sequences. The more recent development of *next generation*, or *deep*, sequencing approaches has increased this to tens of thousands or millions. While this is still a small fraction of the total number of B cells (perhaps $10^9$ in the blood alone [9–11]), it represents a significant, and potentially representative, observational window for circulating B cells. The deep sequencing of BCR repertoires has thus become ubiquitous in the last decade, and is now a key tool in understanding the adaptive immune response.

In order to analyze these data, we typically need to group sequences into *clonal families*, each stemming from a single rearrangement event. We refer to the resulting clusters as a "partition" of the sample, and also use the verb "partitioning" as an alternate term for "clustering" that, while entirely equivalent, emphasizes separating unrelated sequences rather than grouping related ones. The simplest (and still widespread) method is to first group together sequences with the same V and J genes, and then perform single-linkage clustering on the CDR3 sequence with some sequence identity threshold. This method is, unfortunately, often quite inaccurate, especially on repertoires with significant SHM [12]. Two basic reasons for this inaccuracy are that gene calls are often uncertain, and that any single threshold is not optimal for most samples. However a more fundamental issue is that it clusters on observed (mutated) sequences, which allows SHM to spread apart clonal families that should be merged (S2 Fig). A variety of approaches have addressed some of these problems separately, for instance by relaxing the requirement for exact gene calls [13, 14] and using more sophisticated thresholds [15–17]. We have previously shown a substantial accuracy improvement by avoiding all three issues simultaneously [12] using distance-based clustering on inferred naive (rather than observed) sequences, combined with a hidden Markov model (HMM) describing VDJ rearrangement and SHM. By integrating over all possible naive rearrangements, the HMM's inference of clonal relationships remains entirely agnostic to the inherently uncertain inference of naive rearrangement parameters. Some methods also group together sequences with similar inferred mutations, and split apart those with different ones [18, 19]. This can

improve performance for instance on trees with shared mutations on an edge descending from the naive ancestor. However, if there are not enough mutations shared among all sequences then this can incorrectly split lineages apart (S2 Fig).

The performance of all of these clustering methods, however, has been until recently subject to a fundamental limitation. Because the heavy and light chains are found on separate chromosomes, they have typically been sequenced separately, leaving no information on which heavy and light chain sequences pair with each other (which we refer to as "pairing information" or "pair info"). This is clearly an issue if the goal is to synthesize actual antibodies, as potential partners may simply need to be guessed. It also creates a problem of unavoidable overmerging in clonal family clustering when different rearrangement events lead to close to indistinguishable naive sequences. We refer to these as *collisions*, and they can be thought of as involving events whose naive sequences differ by fewer than perhaps 2–4 nucleotides, such that they are practically indistinguishable. While it is not possible to measure the actual frequency of such collisions using unpaired data, we can tell that they are common in light chain because experimentally measured cluster size distributions are vastly more clonal for light than heavy [20]. Note that this unpaired data provides no information on the rate of heavy collisions, and gives only a lower bound on that rate for light. While the theoretical maximum diversity of the heavy chain is large enough to suggest that heavy collisions might be rare, the actual distributions from which parameters are chosen are extremely far from uniform, with for instance the chance of different V gene choices varying by orders of magnitude [3]. This means that the actual collision frequency could be substantial. Note that we can further distinguish between *naive collisions*, in which the naive rearrangements are close to each other; and *mature collisions*, in which the mature sequences from two naive rearrangements have drifted close to each other (S2 Fig).

Fortunately, deep sequencing data with heavy/light pairing information (*paired* data) has become much more common in recent years [7, 21–25]. While previous approaches had low efficiency or cell throughput [26–29], current methods have improved on this. The most widespread method, from 10X Genomics, uses barcoded single cell droplets [24]. Other approaches, however, exist for instance using mass spectrometry [30], physical linking of heavy/light transcripts in droplets [7, 22], or emulsion-based single cell barcoding [23]. There is also the potential for various single cell RNAseq technologies to be applied to B cells [25].

This new paired data provides the opportunity to dramatically improve clustering by using pairing information to refine each single chain partition. For instance if two naive heavy rearrangements are indistinguishable, but their corresponding light rearrangements are chosen independently of each other, the two light families will almost always be easily distinguishable. This procedure is entirely dependent on the extent of this independence, i.e. the extent to which heavy/light pairing is uncorrelated in the underlying biological processes, which we call (a lack of) "biological pairing correlation". For example if indistinguishable heavy events always had exactly the same light partners (perfect correlation), pair info would not contribute to clustering at all. Luckily biological pairing correlations have so far been found to be small [31] or nonexistent [32–34]. While previous work has noted the potential for this approach to clustering improvement [35], there has not been a fully benchmarked implementation that takes advantage of all of the information that paired data has to offer, nor that accounts for the practical quirks of this data.

Ideally, pairing information would consist entirely of pairs of sequences, one heavy and one light. In practice, we encounter imperfect pairing information both from the biological processes such as allelic inclusion described above, as well as from various experimental factors. Sequencing or amplification failure can result in zero partners ("no pairing information") for

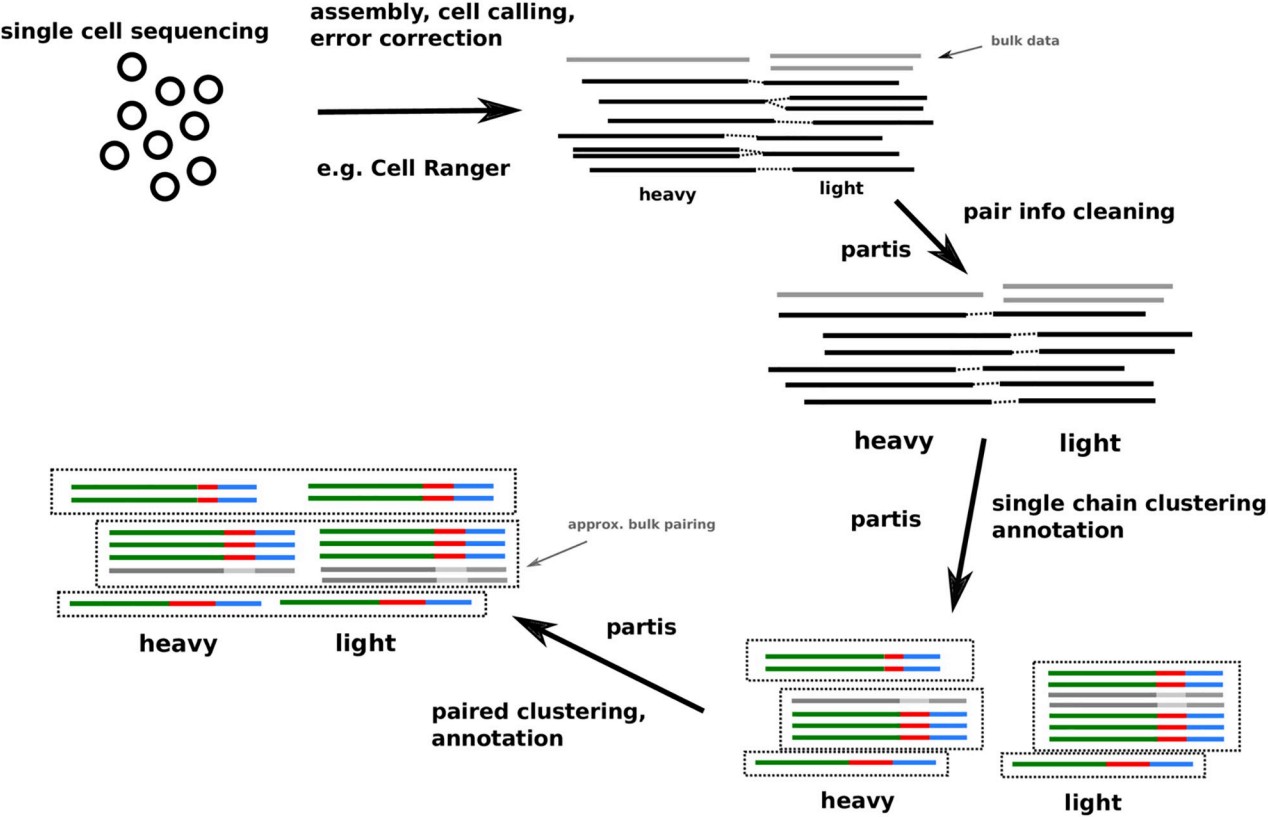

**Fig 1. The `partis` workflow for paired clustering starts with (potentially multiply-) paired sequences, along with possibly some bulk (i.e. non-single cell) data.** It performs clonal family inference on each chain individually, then merges the resulting single chain partitions together into a "joint" or "paired" partition, with the option of approximately pairing bulk data with these clonal families.

some sequences. Also, some fraction of droplets in paired data contain more than one cell, resulting in multiple potential partners ("multiple pairing information").

In this paper we introduce a method for refining single chain partitions using heavy/light pairing information. We compare performance on simulated samples against several single chain methods [17, 36] and two paired methods [19, 37, 38]. We then show that our method's effect on cluster size distributions in real data is very similar to that in matched simulation samples. We also introduce two methods to fix imperfect pairing information: first, a method to disambiguate (or *clean*) pair info in samples that contain more than one cell per droplet; and second, a method to approximately pair (up to clonal family) sequences from a bulk sample that is matched to a single cell (paired) sample. Finally, we describe recent improvements to the single chain methods of the software package into which all of these methods are collected, `partis` (https://github.com/psathyrella/partis). An overview of our workflow is shown in Fig 1.

## Results

### Paired clustering method overview

Our paired clustering method proceeds in two steps: it first infers a partition for each chain individually, then uses the pairing information (or "pair info") to combine them into a single, joint partition of the sequences. There are several options for the first, single chain clustering

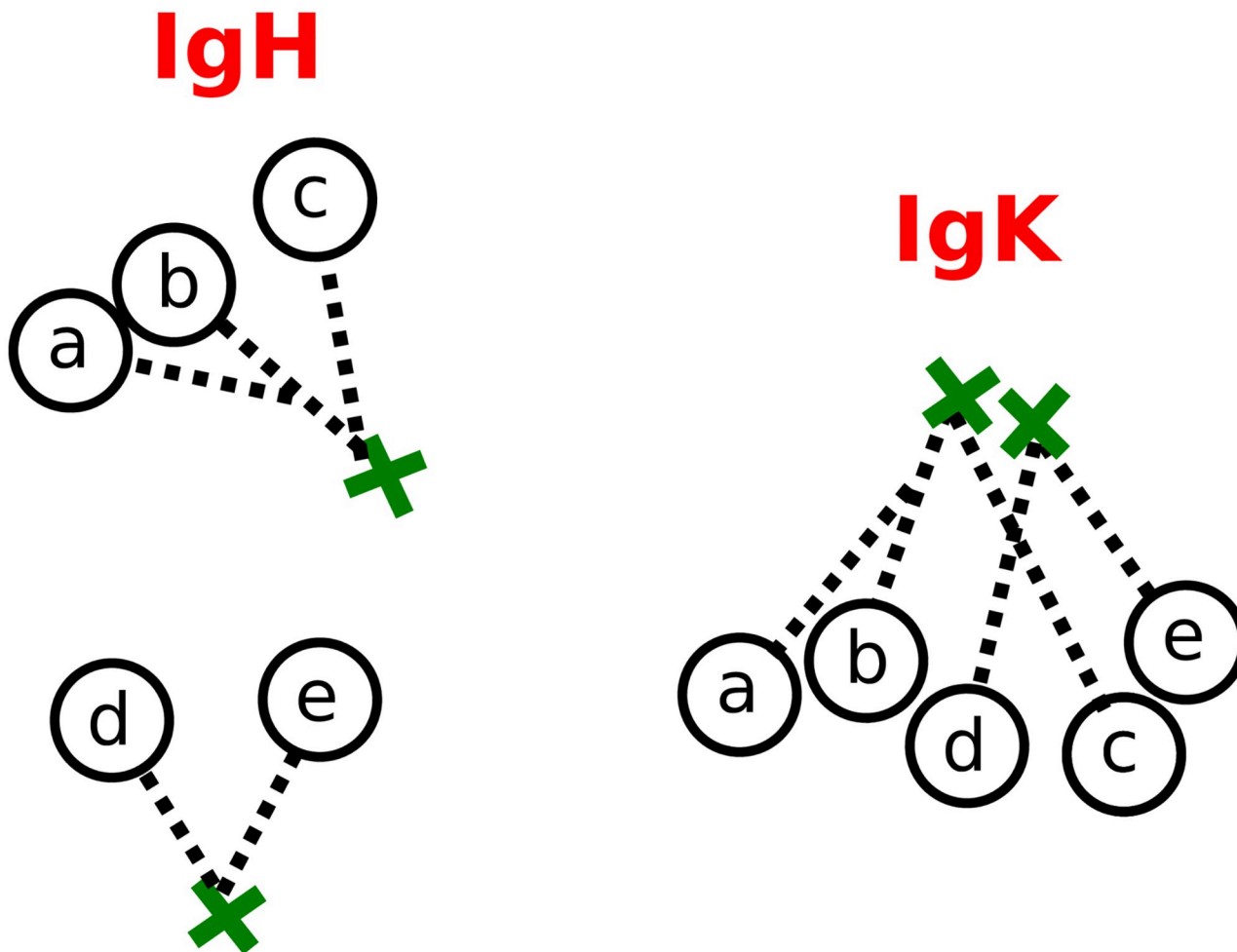

**Fig 2. Schematic representation of our paired clustering method on two families in the heavy chain (left) and light chain (right), with droplet identifiers represented as single letters.** The naive sequences (green crosses) and trees (dashed black lines) represent the collision (i.e. occurrence of very similar VDJ rearrangements) of the two light chain families, while the heavy families are easily distinguishable. The first step of our method groups together apparently-clonal sequences using only single chain information, and would thus merge together the two light chain families. The second step, which refines the single chain partitions using information on which heavy and light chain sequences pair together ("pairing information" or "pair info"), would then use the difference in heavy clusters to split apart the light families.

step, including a variety of tradeoffs between accuracy and speed (see Methods and https://bit.ly/3tuSGjE). The second, paired clustering step refines these single chain partitions, effectively splitting apart clusters in each chain based on information from the other (Fig 2).

## Performance metrics

In order to evaluate performance on simulation, we quantify the extent of overmerging and oversplitting using the usual precision and sensitivity labels for complementary performance attributes. In order to emphasize the importance of optimizing for both at once, we also use their harmonic mean (F1 score). However, it is important to note that there are several different ways to calculate these quantities for clustering performance, and we have chosen only one (see Methods). On real data, since we do not know the correct partition, we instead first verify the similarity of data and matched simulation samples on a variety of parameters, and then compare the results of our clustering method on data and simulation (see Methods).

## Clustering methods

We compare the performance of a variety of different clustering methods in this paper. We show two different methods from our `partis` software package: the full, default method, as well as a faster, more approximate method (*vsearch partis*) that is better suited to extremely large samples, or when quick results are more important than maximum accuracy.

We also show results from several other single chain methods (i.e. that do not use pair info). The simple method of grouping together sequences with the same V and J genes, followed by single-linkage distance-based clustering on CDR3 sequences is still widespread (following [39], we use a threshold of 0.8 in nucleotide distance and call it *VJ CDR3 0.8*). The `SCOPer` package implements spectral clustering with a threshold set by looking for a minimum between two maxima in the sample's pairwise distance distribution [17]. The `MobiLLe` package simultaneously optimizes two criteria, minimizing diversity within families while maximizing that between them [36]. The `SCOPer` package also has a paired option (used in [38]), which we use according to a script written by the authors of the package https://bit.ly/3K8KfBa. We also compare to the official 10X paired clustering method, `enclone` [19]. Note that `enclone` discards sequences that do not have high quality V alignments, which means that it discards many sequences that the other methods do not, particularly those with higher SHM (see Fig 3 legend).

In order to provide intuitive context for the actual methods, we also show two *synthetic* methods, which are generated directly from the simulated true partition. The first, called *synth. 20% singleton*, randomly chooses 20% of sequences from the true partition to split into singleton clusters. This is intended to show a baseline for how a simple operation affects the performance metrics, and is also useful because its performance is similar to the VJ CDR3 0.8 method. The second, called *synth. neighbor 0.03*, merges together families that, with only single chain information, are close to indistinguishable. It does this by merging families whose true naive sequences are closer than 3% in nucleotide Hamming distance. While the exact threshold is arbitrary, it has been chosen such as to provide a useful visual indication of the naive collision frequency.

## Simulation results

The goal of performance evaluation on simulation is to understand behavior in all regions of parameter space. While the entirety of this space for BCR repertoires is extremely complex and high dimensional, it is only important to vary those parameters that have a significant effect on performance. For situations depending on the details of tree shape [40], for instance, this can be quite complex; however the present case of clonal family clustering is much simpler. This is because the difficulty of clustering is mostly determined only by the number of mutations and the naive collision fraction. The normalized number of mutations (SHM fraction) can be thought of as the diameter of each family in rearrangement space (S2 Fig), while the naive collision fraction is determined by the typical distance between naive rearrangements (determined by the locus's inherent diversity and the number of rearrangements).

We show performance for all methods versus SHM fraction in terms of F1 score (Fig 3) and precision and sensitivity (S3 Fig). To show the effect of the paired clustering method itself, we also show performance versus SHM fraction using only single chain information for `partis` (S4 Fig and `SCOPer` (S5 Fig). We note that while the `partis` paired clustering algorithm improves both IgK and IgH (S4 Fig), the paired version of the `SCOPer` method improves IgK while worsening IgH (S5 Fig), the latter of which is driven by a substantial degradation in IgH sensitivity. The deviation of `enclone`'s F1 score from 1 under our simulation conditions, on the other hand, is driven by low sensitivity (i.e. oversplitting) (S3 Fig). This initially surprised

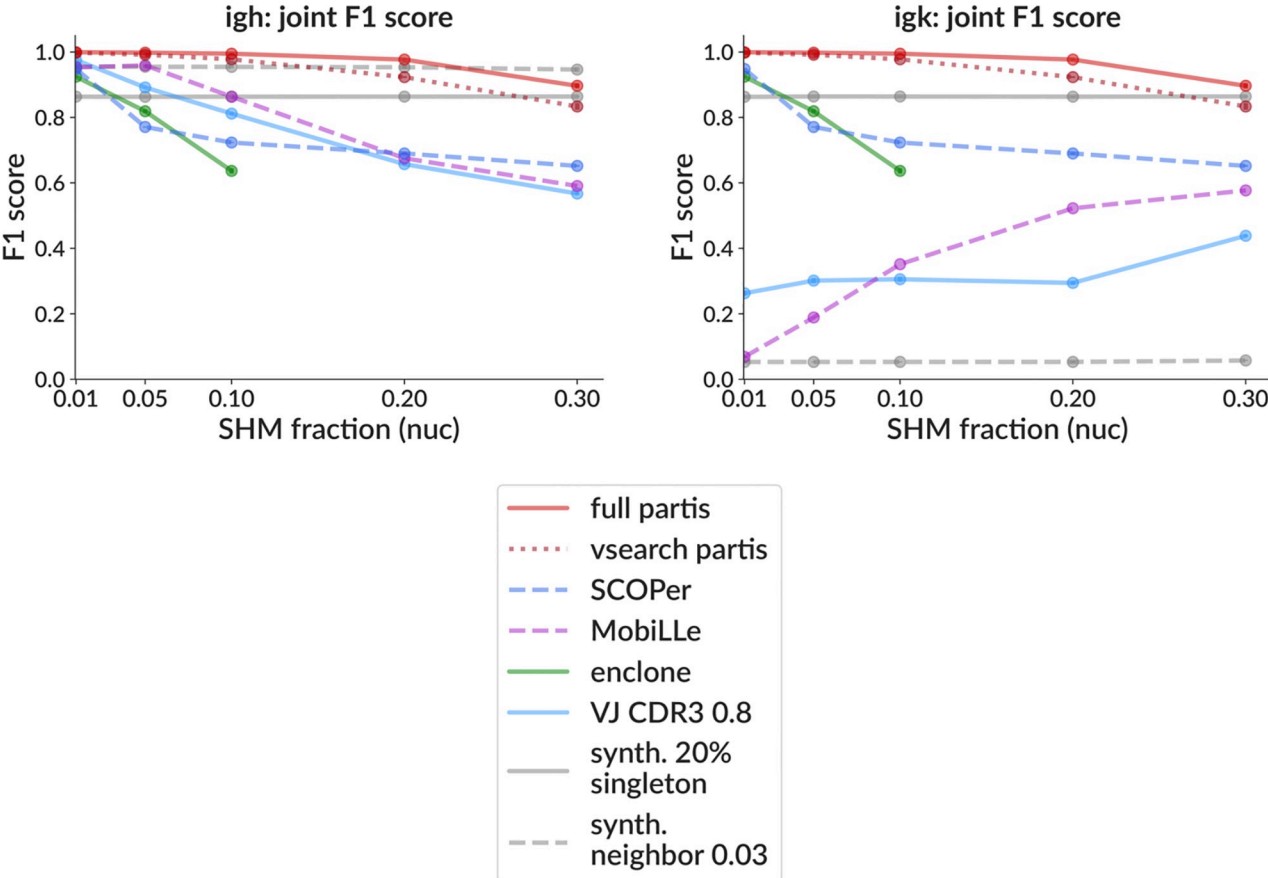

**Fig 3. Clustering performance on simulation as a function of SHM (mean fraction of nucleotides mutated) for heavy chain (left) and light chain (right).** Each point is the mean F1 score (± standard error, often smaller than points) over three samples, each consisting of 10,000 simulated rearrangement events with family sizes drawn from a geometric distribution with mean three. In addition to the inference methods (shown in color), we include two *synthetic* partition methods (grey), which generate incorrect partitions starting from the true partition. These are purely to provide intuitive comparison: the first splits 20% of sequences, chosen at random, into singleton clusters; the second merges together families whose true naive sequences are closer than 3% in Hamming distance ("synth.", see text for details). The F1 score's component precision and sensitivity are plotted in S3 Fig, while the performance with and without using pairing information is compared for partis in S4 and S7 Figs, and SCOPer in S5 Fig. Performance of partis as a function of the number of families (with SHM constant) is shown in S6 and S7 Figs. Note that enclone by design discards some sequences with higher SHM levels, so we display its performance only for those samples where it passes at least 90% of sequences (it discards ≃9% of sequences at 10% SHM, ≃60% at 20%, and ≃94% at 30%).

us, given the results presented in the enclone paper [19]; however further investigation revealed what the performance metrics used in that paper were measuring (S13 Fig).

As described above, besides SHM fraction, clustering difficulty depends largely on the naive collision fraction, which is in turn determined by the locus's inherent diversity and by the number of rearrangements. Since locus diversity is nearly constant for a given species, we next show performance only as a function of the number of rearrangement events (families, S6 and S7 Figs). Note that these parameters largely determine the difficulty of single and paired clustering together: the difficulty particular to the actual paired clustering algorithm is mostly determined by the level of biological pairing correlation, which in this paper we assume is approximately zero.

We also benchmark pair info cleaning. In current 10X data with recommended loading levels, multiple cells only occur in around 1–8% of droplets [41] (although we have encountered a sample with 40%). However, it would be useful to rescue even a small fraction of cells for

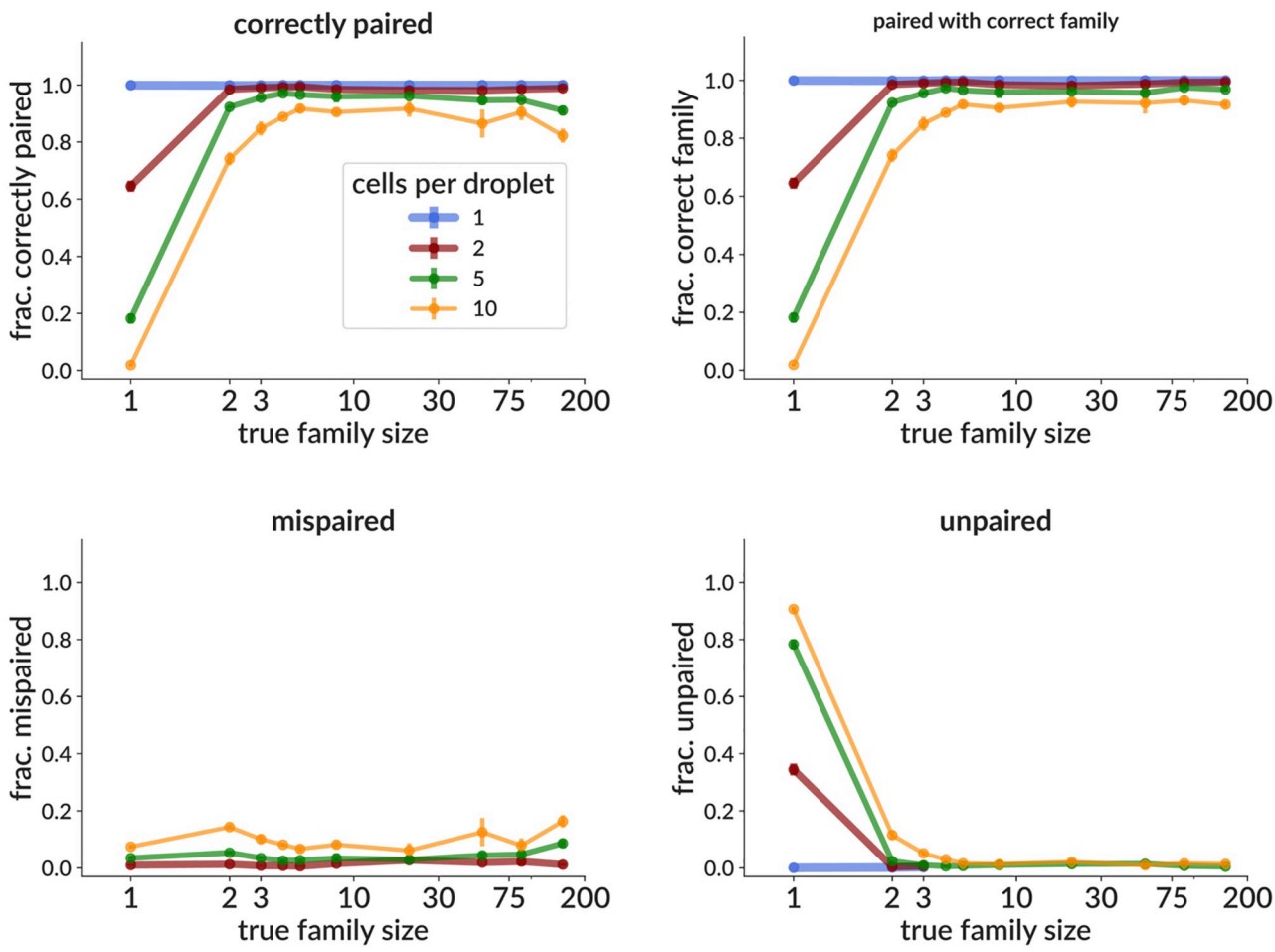

**Fig 4.** Effectiveness on simulation of the pair info cleaning method as a function of true family size, shown as the fraction of sequences correctly paired (top left); and the fraction not correctly paired, split into those mispaired (bottom left) and left unpaired (bottom right). We also show the fraction that are paired with a sequence from the correct family, though not necessarily the correct sequence from that family (top right). Each point is the mean (± standard error, often smaller than points) over three samples, each consisting of 3,000 simulated rearrangement events (around 12,000 total heavy and light sequences). The family sizes are drawn from a distribution inferred from real data. The effects of a variety of cluster size distributions on performance are shown in S8 Fig, in terms of both fraction correctly paired and of final clustering accuracy.

analysis by disambiguating multiple pair info (they are often discarded [42]). Perhaps more importantly, an effective disambiguation algorithm (we call this *pair cleaning*) raises the possibility of purposefully overloading droplets in order to increase the total number of sequenced cells. It would also simplify sample preparation by reducing the necessity for mono-dispersed droplets [43]. Because our method uses clonal relationships to determine correct pairings, sequences in large families are more likely to be corrected than those in small clusters. We thus measure performance on simulation as a function of family size (Fig 4), and see that 1- and 2-sequence families are often left unpaired, especially with many cells per droplet. However in many practical cases, we are most interested in larger families since they are often more likely to be responding to the antigen of interest (for instance a vaccination dose). Thus many experiments that pre-enrich for antigen binding will be biased toward larger families, on which our method performs better. We find for instance that sequences in families larger than 3 are correctly paired 80–85% of the time even with 10 cells per droplet (Fig 4 top left). If we relax this to include sequences that are paired with the wrong sequence, but from the correct family, this

rises to around 90% (Fig 4 top right). We can also visualize performance by comparing overall results (summed over all families in the sample) on samples with different family size distributions (S8 Fig). Here we show both samples with family sizes drawn from a distribution inferred from real data (solid red line; this is about 70% singletons, and corresponds to the samples shown in Fig 4), as well as several samples where all families have the same, indicated size (dashed lines). This shows both the fraction of sequences correctly paired (top), and the effect of this on clustering performance (bottom). It is important to note that, at present, our method does nothing to infer allelic inclusion (i.e. multiple chains per cell).

Because bulk (unpaired) sequencing data can be produced with much greater depth than single cell (paired) samples, it might be possible to increase paired sequencing depth if we could apply pair info from a single cell sample to a matched bulk sample (i.e. a sample drawn from the same pool of cells). We have developed a method that accomplishes this using clonal family information from the merged bulk and single cell samples, such that bulk sequences in families with at least one paired cell will be paired with sequences from the correct (or at least similar) corresponding opposite-chain family. We include "similar" because these are families from the single chain partition (since we don't yet have pair info), so will include collisions between different, largely indistinguishable, events. Such collisions can result in choosing a partner from the wrong family, but because the two families are very similar, the pairing may still be functional. We measure the performance of this method both vs. true family size (since larger families are more likely to contain a paired sequence), and vs. the fraction of the merged sample derived from bulk data; and report the fraction paired with both correct and similar families (Fig 5). We find, for instance, that with a bulk data fraction of 0.8 (i.e. if our total sequencing depth is five times our single cell sample), around 90% of sequences in families of 10 or larger are paired with a sequence from the correct family (Fig 5 left), and >99% are paired with a similar family (Fig 5 right). We also show the fraction correctly paired (i.e. paired with exactly the correct sequence), mispaired, and unpaired (S9 Fig). Note that sequences that are not paired with a similar family are essentially all left unpaired, i.e. very few sequences are paired with a family significantly different from their correct partner. These unpaired sequences are from families that did not happen to contain any single cell (paired) sequences.

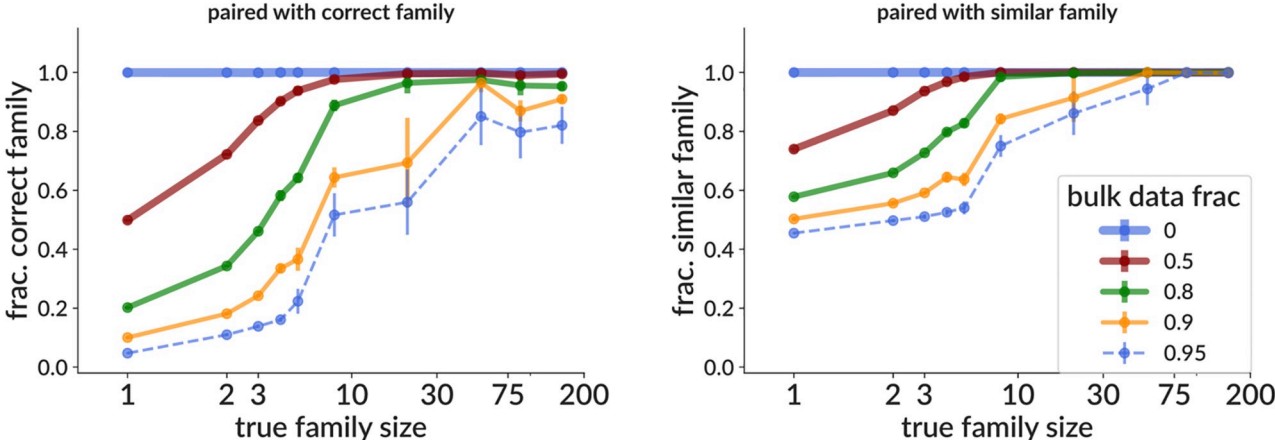

**Fig 5. Effectiveness on simulation of the approximate bulk data pairing method as a function of true family size, shown as the fraction of sequences paired with a sequence from the correct family (but not necessarily the correct sequence, left) and the fraction paired with a family similar to the correct family (right).** The method merges together a single cell and a bulk sample drawn from the same pool of B cells, and the "bulk data fraction" shows the fraction of this merged sample that stems from the bulk sample. "Similar" families are defined as families with true naive sequences separated by nucleotide Hamming distances of 3 or less. The fraction of sequences correctly paired, mispaired, and left unpaired are shown in S9 Fig. Other details same as in Fig 4.

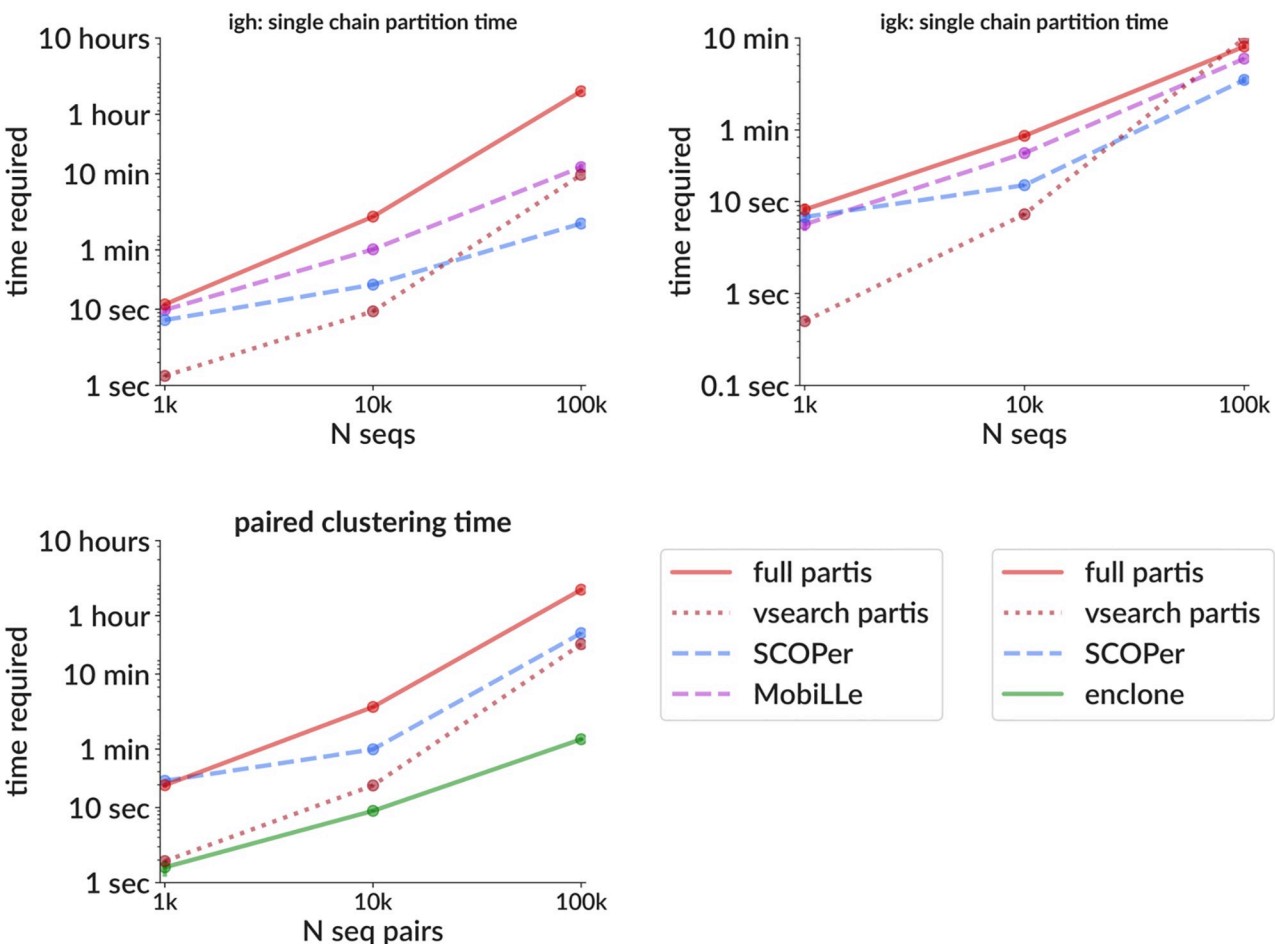

**Fig 6. Time required for clustering for a variety of methods for both single chain clustering (top, for IgH [left] and IgK [right]) and paired clustering (including single chain clustering, bottom left).** Note that for `partis` the actual paired clustering takes only around five minutes on the 100k samples, so for `partis` the bottom plot time is roughly equal to the sum of the top two plots (at the moment the single chain clustering for heavy and light are not run concurrently). Each point is the mean (± standard error, often smaller than points) over two samples with the indicated size, run on a single desktop with a 14-core Intel i-99940X processor and 128GB memory (maximum memory usage for `partis` on the 100k samples was around 9GB). The size of each family is drawn from a geometric distribution with mean 10. Note that because `enclone` by design discards sequences with high SHM, here it is clustering only the ≃80% of sequences that it passes.

Note also that by construction, the fraction actually correctly paired is simply the single cell (non-bulk) fraction of the sample (S9 Fig). Note also that these fractions are calculated for each individual heavy and light sequence and then averaged. This is designed to represent the "typical" experience of an initial heavy/light sequence pair, but note that each of these fractions can on average be different for heavy chains vs. light chains. This latter point means, for instance, that these numbers cannot be directly compared to [35], which calculates them for one chain only. Finally, it is important to note that the only way to be certain that an incorrect pairing (even within the correct family) results in a functional antibody is to actually synthesize it.

We also measure the time required for each method (Fig 6). While bulk BCR sequencing samples can contain millions of sequences, at the moment paired data typically has only 1,000 to 10,000. This means that even the slowest methods finish in minutes on typical paired data. One recent exception [44], however, made a sample with 3.4 million sequences in 1.6 million droplets; the faster `vsearch partis` mode partitions this sample in about 20 hours, with a

maximum memory usage of 60GB. For single chain clustering there is a rough tradeoff between accuracy and time required. The `partis` implementation of paired clustering using these single chain clusters, however, does not take a significant amount of time (a few seconds on 10,000 sequence pairs). We do not show memory usage, since we do not find it to be a limiting factor (for instance the 10,000 sequence paired samples used less than around a GB).

## Real data results

We show clustering results on real data in the form of cluster size distributions for single and joint partitions. These are shown for a sample from the 10X website [45] together with results on a parameter-matched simulation sample (Fig 7), and similarly for three other samples in https://doi.org/10.5281/zenodo.5860143. The extent of splitting, i.e. the difference between the single and joint distributions in either data or simulation, tells us the paired clustering method's determination of the extent of single-chain overmerging.

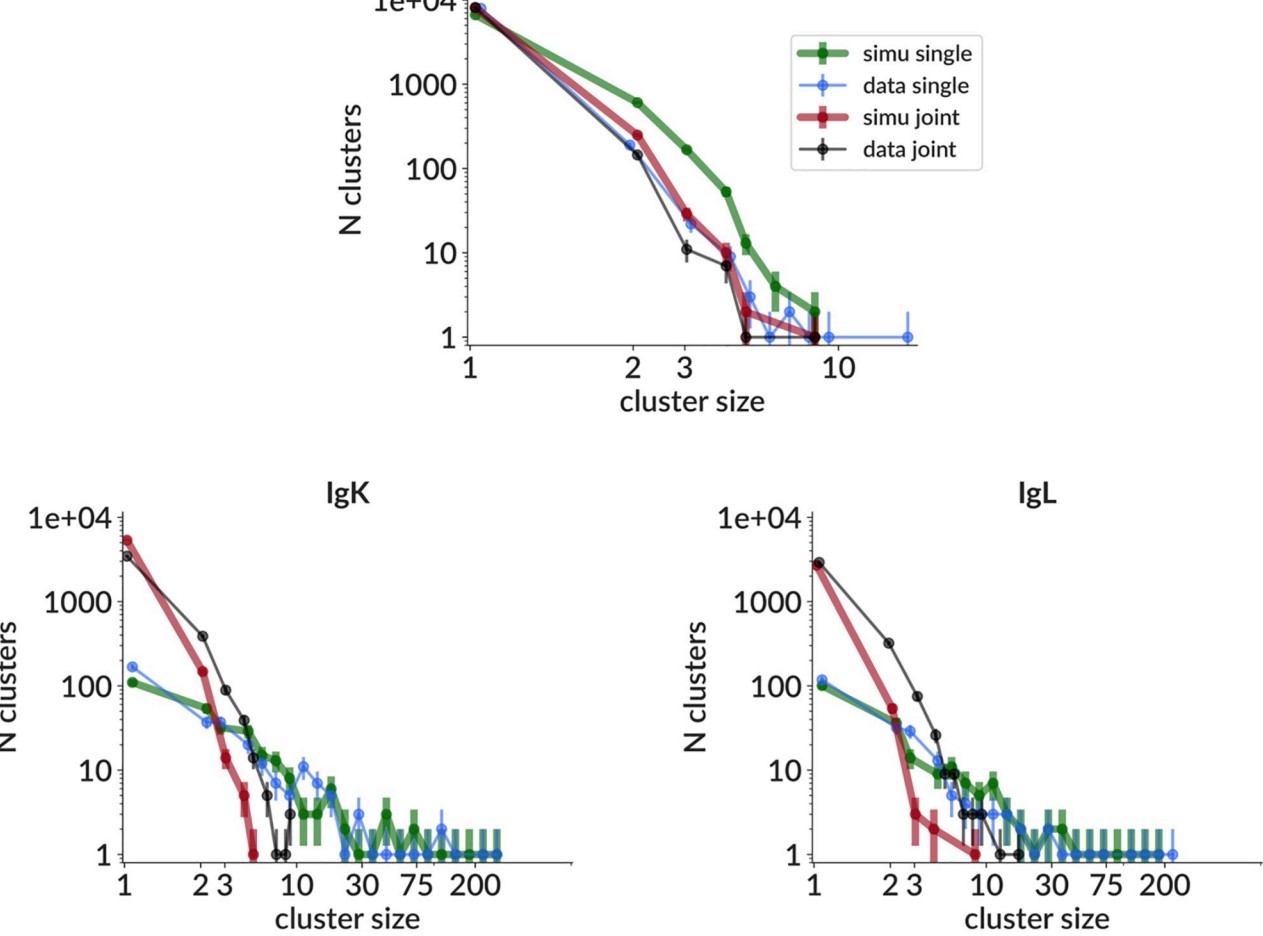

**Fig 7. Cluster size distributions on real data (thin lines) and simulation (thick lines) for single chain (no paired clustering, blue/green) and joint (with paired clustering, black/red) partitions for IgH (top), IgK (bottom left) and IgL (bottom right).** The real data sample is from the 10X website [45], while the simulation sample is generated to match that particular real data sample, i.e. from parameters inferred from that real data sample. Direct comparisons of other parameter distributions for these two samples are shown for a handful of parameters in S10 Fig. Comparisons for all parameters, along with similar analyses for three other real data samples from [45] may also be found at https://doi.org/10.5281/zenodo.5860143.

Furthermore, the similarity of the data and simulation distributions for both single and joint partitions gives us confidence that both the single chain simulation, and the uncorrelated pairing assumption, suitably approximate the underlying processes for the purposes of benchmarking. The discrepancies between data and simulation in IgH are partly the result of small sample size: this particular simulation sample did not recapitulate the very tail of the data distribution (i.e. the single data cluster of size 18), but if we had drawn many such samples they (by construction) would have sampled the very tail. For light chain, on the other hand, the single chain distributions are well matched, but the simulation joint distribution is less clonal than that in data. It is possible that this reflects a breakdown of the assumption of no biological pairing correlation: low (but non-zero) correlation in data would manifest as having split fewer than expected light clusters based on heavy information (and vice versa). Overall, because these discrepancies are much smaller than the difference between the light chain single and joint distributions, which control the difficulty of inference, we think this level of agreement is sufficient for our current benchmarking purposes. In the future we plan to investigate biological pairing correlation in data, at which point we will be able to more thoroughly investigate this. We also directly compare distributions for a handful of other parameters on data and simulation for this sample (S10 Fig, with all other parameters and samples in https://doi.org/10.5281/zenodo.5860143).

We also show the results of our pair cleaning algorithm on two real data samples (Fig 8). The first, demonstrating typical (low) levels of multiple pairing information (i.e. multiple cells per droplet), corresponds to an example dataset from the 10X website [45]. The second, on the other hand, is a 10X sample from our collaborators where less than half of sequences start out uniquely paired (not yet published, but raw data included in https://doi.org/10.5281/zenodo.5860143). Our method leaves both samples with the majority of sequences uniquely paired, while a small fraction are left unpaired.

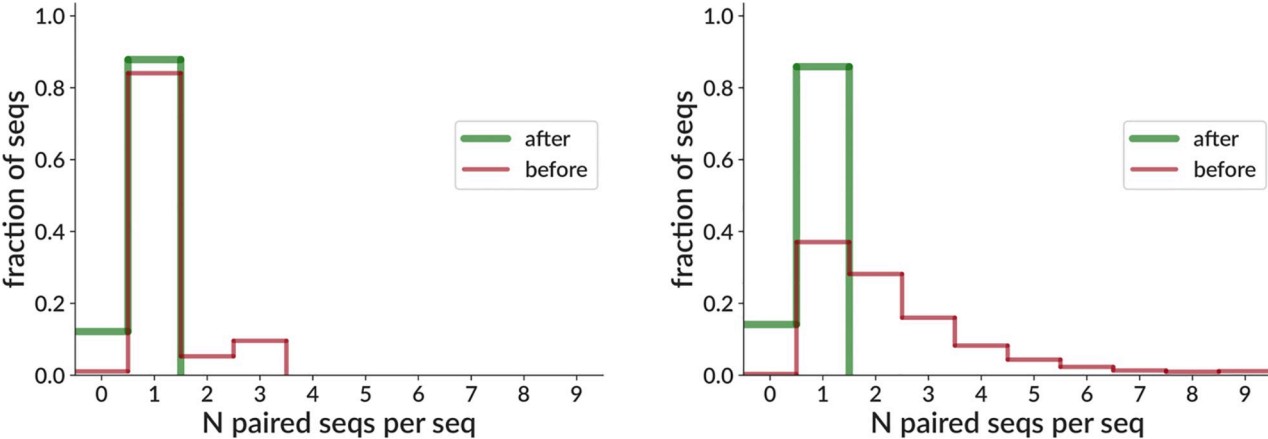

**Fig 8. Pair info cleaning effectiveness on real data for a relatively well-paired sample (left, same sample as Fig 7) and a sample with substantial numbers of multiply-paired sequences (right, not yet published, but raw data included in https://doi.org/10.5281/zenodo.5860143).** The x axis shows the number of sequences paired with each sequence ("paired seqs per seq") before and after application of our pair info cleaning algorithm. Thus for example a perfectly paired sample (with perfect allelic exclusion, no dropout, etc.) would have all sequences in the 1-bin (i.e. uniquely paired), while a sample with two cells in each droplet would have all sequences in the 3-bin. So an ideal application of our algorithm would leave every sequence uniquely paired (all in the 1-bin), but in practice some are left unpaired (0-bin).

## General `partis` improvements

While many improvements have been incorporated into the `partis` methods since initial publication [46], recent changes to the inference of the collection of parameters necessary to specify a naive rearrangement (gene calls, insertions, and deletion lengths), which we refer to as an *annotation*, are particularly significant and will be described here. In order to handle large data sets, the emission probabilities in the multi-HMM are calculated independently, i.e. the evolution of each family is viewed as a star tree for annotation purposes. This is of course a poor approximation for families that have highly imbalanced trees (those with large variance in root-to-tip distances). In order to address this, we have introduced an iterative *subcluster annotation* scheme (see Methods) that maintains the speed of the star tree assumption, but largely ameliorates the inaccuracy in annotation (Fig 9). Here we compare to several methods: `linearham` [47], which uses a Bayesian phylogenetic HMM to fully calculate the posterior probabilities of different trees and annotations; and `IgBLAST`, which is one of the most popular general annotation methods. The current default "full" `partis` comes closer to replicating `linearham`'s accuracy while running much more quickly (`linearham` typically takes minutes to hours on single families). It is important to note, however, that `linearham` provides vastly more (and better) information in its posterior probabilities than are in `partis`'s largely heuristic uncertainty estimates, so is much better suited to detailed analysis of single families.

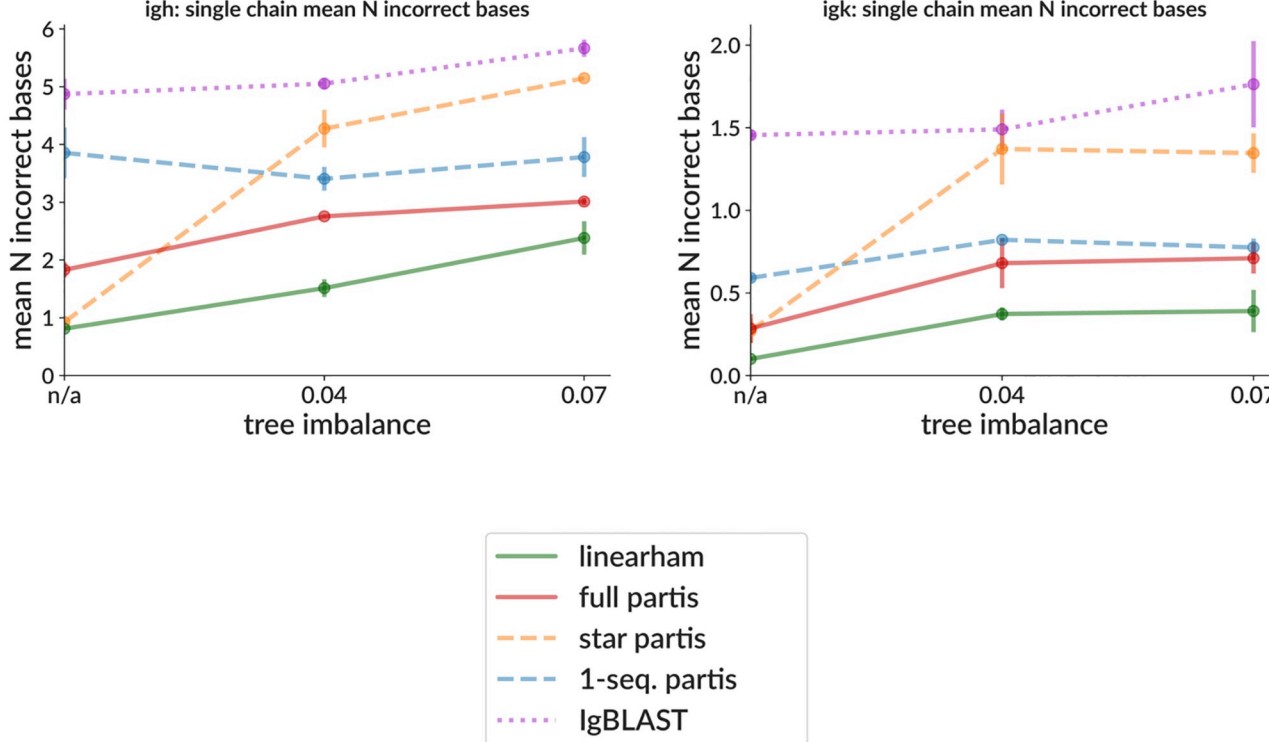

**Fig 9. Naive sequence inference accuracy on simulation as a function of tree imbalance.** Accuracy is measured as the Hamming distance separating the true and inferred naive sequences. Tree imbalance (the standard deviation of root-to-tip distances) is shown for samples with precisely zero imbalance ("n/a", since it uses a different simulation framework), and samples generated from the imbalanced trees from [47]. Several `partis` versions are shown: "1-seq." uses only information from each single sequence for its annotation (rather than from all family members); "star" approximates each family as a star tree (as in `partis` versions before May 2021 https://bit.ly/3KIjFiF); "full" is the current default, including subcluster annotation, which improves on the star tree assumption by iteratively annotating small subtrees. Each point is the mean (± standard error) over two samples with 15% mean nucleotide SHM, each consisting of 50 simulated rearrangement events with sizes of around 50 (drawn from geometric for zero imbalance; otherwise taken from the trees from [47]).

## Discussion

We have introduced methods to enable inference of B cell clonal families on samples with heavy/light chain pairing information. We first showed a method to refine single chain partitions using pairing information. This method dramatically improves accuracy on simulation compared a variety of other single chain methods, and gives compatible results on real data. We then showed two methods that use inferred clonal family information to fix initially imperfect pairing information. The first method disambiguates (or *cleans*) pair info in samples that contain more than one cell per droplet, which could simplify sample preparation by reducing the need for mono-dispersed droplets, as well as allowing overloading of droplets to increase sequencing depth. This method can salvage pair info for around 90% of sequences in families larger than three even in simulation samples with 5–10 cells per droplet; families smaller than this are more frequently left unpaired (especially singletons), while typically a few percent of sequences are mispaired. The second method uses pair info from a single cell sample to approximately pair (correct up to family) sequences in a matched bulk sample (i.e. drawn from the same pool of B cells). Finally, we showed that our new subcluster annotation scheme improves annotation accuracy on families with imbalanced phylogenetic trees.

Some previous experimental papers have used various methods with non-public implementations to cluster paired data. For instance [48] grouped together sequence pairs with identical heavy and light V and J genes, and identical CDR3 amino acid sequences. Two other recent papers similarly grouped together identical heavy and light V and J genes and CDR3 lengths, but then clustered with `SCOPer`'s hierarchical method on heavy chain only [49, 50].

During the revision stage of this paper, the official 10X paired clustering method, `enclone`, was described in a preprint [19]. This method is not open source (it is available only as binaries), but is widely used on 10X data. Its paired clustering algorithm consists of an extensive series of heuristic merging steps using criteria such as shared mutations, V region SHM indels, mutation imbalance between different regions, and CDR3 nucleotide identity.

Another study used early 10X data to investigate the potential for improving heavy partitions with light chain information [35]. Using four samples with sizes of 1,000–10,000, the authors performed single chain `SCOPer` clustering on heavy chain sequences, then looked for heavy clusters consisting of sequences whose light chain partners were "inconsistent", defined as different V gene, J gene, or CDR3 length. Since inference of these quantities is quite accurate, such clusters likely represent heavy chain families that were erroneously merged, and thus cases where light chain information could improve clustering. They found the fraction of such inconsistent clusters to be between 3 and 17% on the four samples, which is roughly in line with the collision frequency that we see in simulation (S6 Fig top left, difference between grey dashed line and 1, N families between 1k and 10k). It is important to note, however, that while the collision frequency is small for small samples, it increases very rapidly with the number of families, i.e. light chain information is much more important for larger samples. Another important point is that it is not possible to determine how many of these inconsistent clusters stem from actual collisions, as opposed to simply inaccurate or suboptimal clustering. The ideas from [35] have been implemented in `SCOPer` and used in practice [38].

Prior work in the vein of our pair info cleaning, as far as we are aware, consists of custom approaches applied in experimentally-driven papers aimed at addressing multiple chains per cell (i.e. allelic inclusion). In this paper, on the other hand, we focus on creating a method for handling multiple cells per droplet (although in future versions we plan to address multiple chains per cell). In [51], for instance, only the heavy chain sequence with the most unique molecular identifiers (UMIs) is retained from any droplets with multiple heavy chains. In [52],

on the other hand, IgK reads are discarded from any droplets that have both IgK and IgL, since if both are from the same cell, the IgL will only have rearranged upon failure of IgK.

It is also possible to use experimental techniques to allow droplet overloading, for instance by pooling samples with distinguishable genotypes [53], or by adding sample-specific barcodes to surface-protein-binding antibodies [54]. We note, however, that without a computational disambiguation scheme, these methods would likely still be limited to only one cell per droplet from each subject.

There have also been experimental papers that use individual families in single sequence data to look for similar sequences in a matched bulk sample, for instance [50]. This addresses the same need as our approximate bulk pairing, however our method provides a codified, repeatable method together with computational validation. Our method also operates comprehensively on the entire repertoire at once rather than focusing on a single target sequence at a time.

There is room for improvement in our paired clustering method. The inference of clonal families with pairing information would be more intuitively straightforward, and likely also more accurate, if we used pairing information from the start. In other words, when preclustering with Hamming distance on inferred naive sequences, we could use the distance over both heavy and light chain sequences. Then, in the likelihood-based HMM clustering, we could at each step simply take the product of likelihoods over both chains. However, this would require rewriting tens of thousands of lines of C++ and Python code to operate on two simultaneous rearrangement events (heavy and light), rather than one at a time, while still maintaining the existing single-chain functionality.

The effectiveness of our pair info cleaning method suggests that it may be possible to purposefully overload droplets in order to achieve higher sequencing depth. This idea is quite similar to that explored with the pairSEQ algorithm for T cells [55]. The difference for B cells, compared to T cells, is that clonal assignment is not trivial; however with `partis` the assignment is accurate enough that the approach appears feasible. We also note that we have used the pair cleaning algorithm with good results on (not-yet-published) non-10X data that flow-sorted cells into wells, but left multiple cells in some wells. Our method may also allow for simplified sample preparation, since the Poisson statistics determining cell-bead co-encapsulation often require very dilute suspensions to achieve mono-dispersion, which compromises efficiency [43].

Our pair cleaning method could be improved by addressing multiple chains per cell, for instance by distinguishing multiply-paired heavy chains from multiple cells in a droplet. As suggested in [21], the latter would result in apparent cross-pairing of all pairs, while the former would show only limited combinations.

Our approximate bulk sample pairing method suggests that paired sequencing depth could also be increased by using matched single cell and bulk samples drawn from the same pool of cells (e.g. a single blood draw). In contrast to pair info cleaning (disambiguating multiple cells per droplet), this approximate bulk pairing comes with the substantial caveat that we are seeking only to pair sequences with the correct family, not the correct sequence. For detailed phylogenetic studies, for instance, this will often not be sufficient. In other cases, however, we are simply searching for functional antibodies, and many incorrect sequences from the correct family will likely work. It is also worth noting that we can probably apply both approaches together: overload droplets in a single cell sample to get more real paired sequences, and then combine this with a bulk sample which we can then approximately pair. Our approximate bulk pairing method would also benefit from evaluation on real data samples; here we have only tested it on simulation.

Another potential future direction is the investigation of biological pairing correlations. Earlier work in this area with small sample sizes found no evidence for biological pairing correlation, using single cell PCR of genomic DNA from 144 IgM cells [32], and 365 IgG cells [33]. Later work with larger samples, however, detected pairing preferences for a small proportion of germline genes using 545 human and 1,456 mouse antibodies from the KabatMan database [31]. Evidence for the coevolution of particular non-self-reactive heavy/light gene combinations [3] provides additional, independent support for some level of preference. Work on T cells, meanwhile, while not necessarily informative for B cells, has found that heavy/light chain pairing is almost, but not quite, uniformly random [56].

## Methods

### Reproduction

All inputs and outputs can be found at https://doi.org/10.5281/zenodo.5860143; however any part of the analysis can also be rerun, for instance with modified parameters, using https://bit. ly/3Pjll4P.

### Paired clustering algorithm

The goal of clonal family inference is to construct a partition of the sample by grouping together the sequences stemming from each single rearrangement event. Our paired clustering method does this in two basic steps: it first clusters heavy and light chains individually with the `partis` single chain partition method, then refines these *single chain partitions* using their mutual pairing information. While we will give an overview of the single chain method, we refer to the original publication for details [12].

**Single chain clustering.** `partis` single chain clustering begins by collapsing together all sequences with identical inferred naive sequences. We will use *cluster* to mean a set of sequences together with its ordering. Here, and at each later stage, upon forming a new cluster we immediately infer the annotation (the collection of gene calls, insertions, and deletion lengths necessary to specify a naive rearrangement) of that cluster. This is because annotation inference on several simultaneous sequences is much more accurate than on single sequences (see Key Translation below).

Single chain clustering then proceeds via hierarchical agglomeration. This consists of considering each pair of clusters in the current partition and merging the pair deemed most likely to be clonal, using a likelihood calculated with a multi-HMM describing that cluster of sequences. This is repeated until none of the pairs are more likely to be clonal than non-clonal. At each step, we ignore cluster pairs whose inferred naive sequences are very different, and immediately merge those that are very similar, using per-sample thresholds that vary with mean SHM level, and that were originally calculated with a simulation-based optimization procedure [12]. We thus only calculate the HMM likelihood for cluster pairs with somewhat similar (but not too similar) naive sequences. The multi-HMM lets us calculate separately both the likelihood that two sequences stem from the same rearrangement, and that each of them stem from separate events. We merge pairs for which the first is larger, and stop clustering when no pairs meet this criterion (or, optionally, continue beyond this point to explore less-likely partitions). The end result is thus not just a single partition, but a list of partitions together with their likelihoods.

The vsearch `partis` method similarly begins by grouping together sequences with identical inferred naive sequences. Each such group is then represented by its common naive sequence, and these are then grouped by CDR3 length. Each such CDR3 length class is then passed to the `vsearch` clustering method [57], which performs highly optimized distance-

based clustering using a specified threshold. We use a threshold chosen with a simulation optimization procedure similar to, but separate from, that described above for naive Hamming preclustering.

**Refinement with pairing information.** At this point we have partitions for the heavy and light chains that each contain sequences from the same cells. We want to resolve the discrepancies between these two partitions. Most commonly, a single cluster in one partition will correspond to multiple clusters in the other, and these multiple clusters will usually have very different annotations (e.g. different CDR3 lengths). While we handle the more general case of arbitrarily discrepant clusters, this common case leads to the choice of a fundamental bias in our approach: when resolving discrepancies we look only to split clusters, not merge them. Another way to look at this is that information from the other chain can give us an extremely strong signal that we should split a cluster (if the multiple clusters are very different), but on the other hand given multiple clusters in the current chain, having very similar sequences in the other chain is not a clear signal that we should merge them. Thus the net effect of our method is almost always to split apart single chain clusters; however because of the need to consider all clusters at once, this splitting is not accomplished directly, but instead by avoiding specific merges.

We now introduce the algorithm more formally. For this, the underlying key objects are pairs of heavy and light chain sequences and clusters thereof. We can think about these pairs as actual pairs of sequences, or as an identifier that is shared between a heavy chain sequence and a light chain sequence, but is unique among such pairs. These identifiers correspond to the labels in (Fig 10). Assume we are given a partition $P_h$ on the pairs/identifiers based on

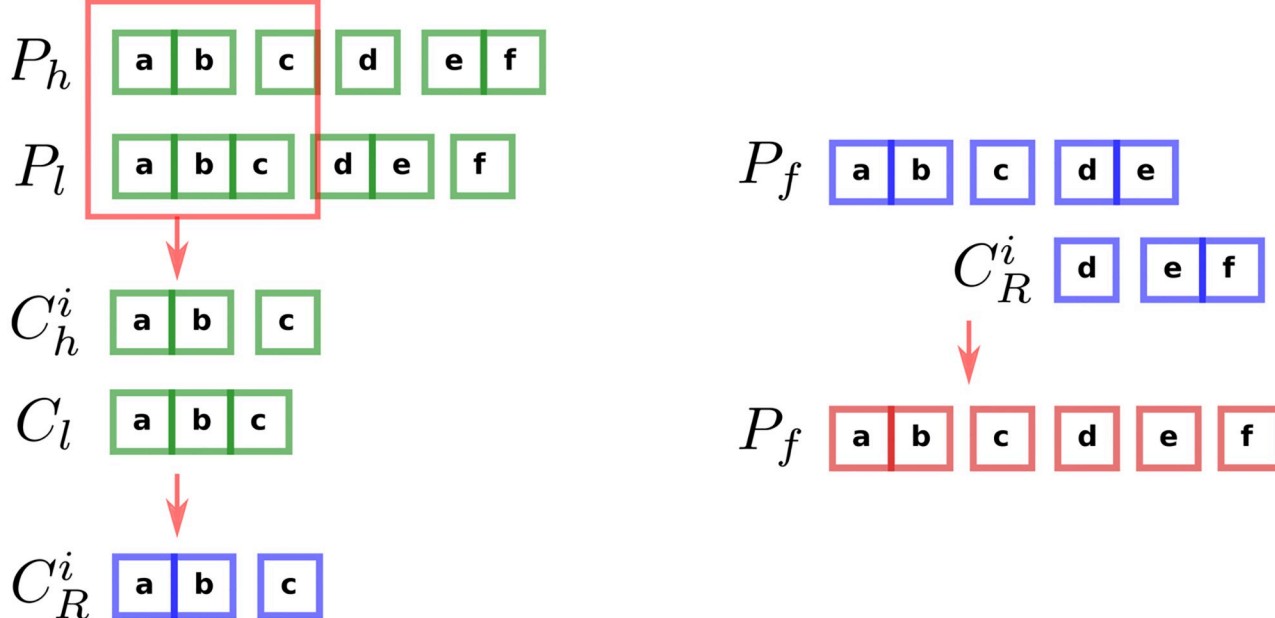

**Fig 10. Resolving discrepancies for one cluster between the heavy and light partitions (left) and incorporating a (different) group of resolved clusters into the final partition (right).** Left: After selecting a single cluster $C_l$ from one partition, we then take all clusters from the other partition (the $\{C_h^i\}$) that contain any of the same three sequences **a**, **b**, and **c**. We then determine how $C_l$ should be split apart using the $\{C_h^i\}$, and finish with the resulting "resolved" clusters $C_R^i$. Right: After several rounds of discrepancy resolution, we have a partially complete final partition $P_f$ into which we must incorporate each new group of resolved clusters $C_R^i$. We do this by removing common sequences from the larger of any two clusters that share sequences (regardless of whether the larger cluster is in $P_f$ or in $C_R^i$). In this case, this means we remove the **e** from **de** in $P_f$, as well as removing the **f** from **ef** in $C_R^i$.

heavy chain sequences and light chain partition on the pairs/identifiers $P_l$ based on light chain sequences.

We begin by considering, in turn, each cluster in both the heavy partition $P_h$ and light partition $P_l$ (see Algorithm 1). Consider a cluster $C_l$ from $P_l$: its size and constituent light chain sequences have been determined during single chain clustering by the likelihood that they are clonally related. We will walk through the algorithm to completion using $C_l$. We now want to see if there might be convincing evidence in $P_h$ that we should split $C_l$. We thus proceed by collecting the list of clusters from $P_h$ that overlap with $C_l$, which we call $\{C_h^i\}$ (Fig 10).

In order to determine if and how to split apart $C_l$ using the $\{C_h^i\}$, we first separate the $\{C_h^i\}$ into groups by CDR3 length. Since inferred CDR3 length is essentially always correct, we definitely want to split $C_l$ apart into sets if the corresponding heavy chain sequences have a different CDR3 length. Within each such CDR3 group, we also want to split clusters that have different enough inferred naive sequences that they confidently come from different rearrangements. For each cluster in the CDR3 group, we thus make a list of clusters from which it should be split (its "split list"): those with inferred naive Hamming distance above some threshold. This threshold is currently that described for *vsearch partis* above, and depends on the sample's SHM level, but is typically around 5%. Future optimization of this threshold may prove worthwhile.

The algorithm implements the splitting in terms of "split lists" for each cluster that contain other clusters with which it should *not* be merged. The splitting procedure described above is then actually implemented by merging together all clusters that do not appear in each other's split lists (Fig 10). This effectively preserves all the between-cluster boundaries in each partition of which we are relatively confident, while merging everything else.

We have now "resolved" the discrepancies between $C_l$ and the $\{C_h^i\}$, and thus call the resulting clusters the resolved clusters $C_R^i$. Note that $C_l$ and the $\{C_h^i\}$ do not in general contain identical sequences.

We must now incorporate the resolved $C_R^i$ into the final partition $P_f$ (Fig 10). Since the $C_R^i$ constitute a sub-partition (i.e. have no duplicate sequences), adding the first group of them, when $P_f$ is empty, is trivial. As we proceed, however, and $P_f$ grows, their incorporation becomes increasingly complex, since the $\{C_h^i\}$ (and thus $C_R^i$) for one $C_l$ can in general overlap in a complicated way with the $\{C_h^i\}$ for another $C_l$. In other words, when we finish a new group of $C_R^i$, we must go through the clusters already in $P_f$ with which these $C_R^i$ overlap, and decide whether to believe the splits between existing clusters in $P_f$, or the new ones in $C_R^i$. Since the goal of the discrepancy resolution procedure from which we derive the $C_R^i$ is to introduce new splits about which we are highly confident, and because a new split is likely based on new information, we do this largely by believing the more-highly-split alternative. To accomplish this, we examine each pair of clusters (one from the $C_R^i$ and one from $P_f$) that share sequences. If this is the first such pair, we simply remove and discard the common sequences from whichever of the clusters is larger. For subsequent pairs, in order to avoid merging sequences from different resolved clusters, we must remove the common sequences from both, and add the common sequences as their own cluster.

We have just resolved the discrepancies between one cluster $C_l$ in $P_l$ and all clusters in $C_h$ with which it overlaps. We then repeat this procedure for every other cluster in $P_l$, and similarly every cluster in $P_h$. The final result is then a single partition that we use for both heavy and light chain sequences.

## Pair info cleaning

Ideally, the pairing information for a sequence would always consist of a single unique paired sequence with reciprocal pairing information. In practice, however, due either to failed

sequencing of one chain, the presence of multiple cells per droplet, or a combination of the two, we can also get zero or more than one paired sequence. Because our paired clustering algorithm requires that all sequences have precisely one partner, we have developed a method to infer which of a number of potential partners is correct. In cases where it cannot, it removes all pairing information from the sequence. The result is thus that after application all sequences have either zero or one partner.

Our method is based on the observation that correct pair info will be shared among members of a clonal family, while spurious partners will not. This depends on the fact that the processes that apportion cells to droplets are entirely independent of clonal family identity. If we consider a single cluster $C_h$ in the heavy chain partition, the correct partner for each sequence will be in the same family as the correct partners of other sequences in $C_h$ (Fig 11). For this

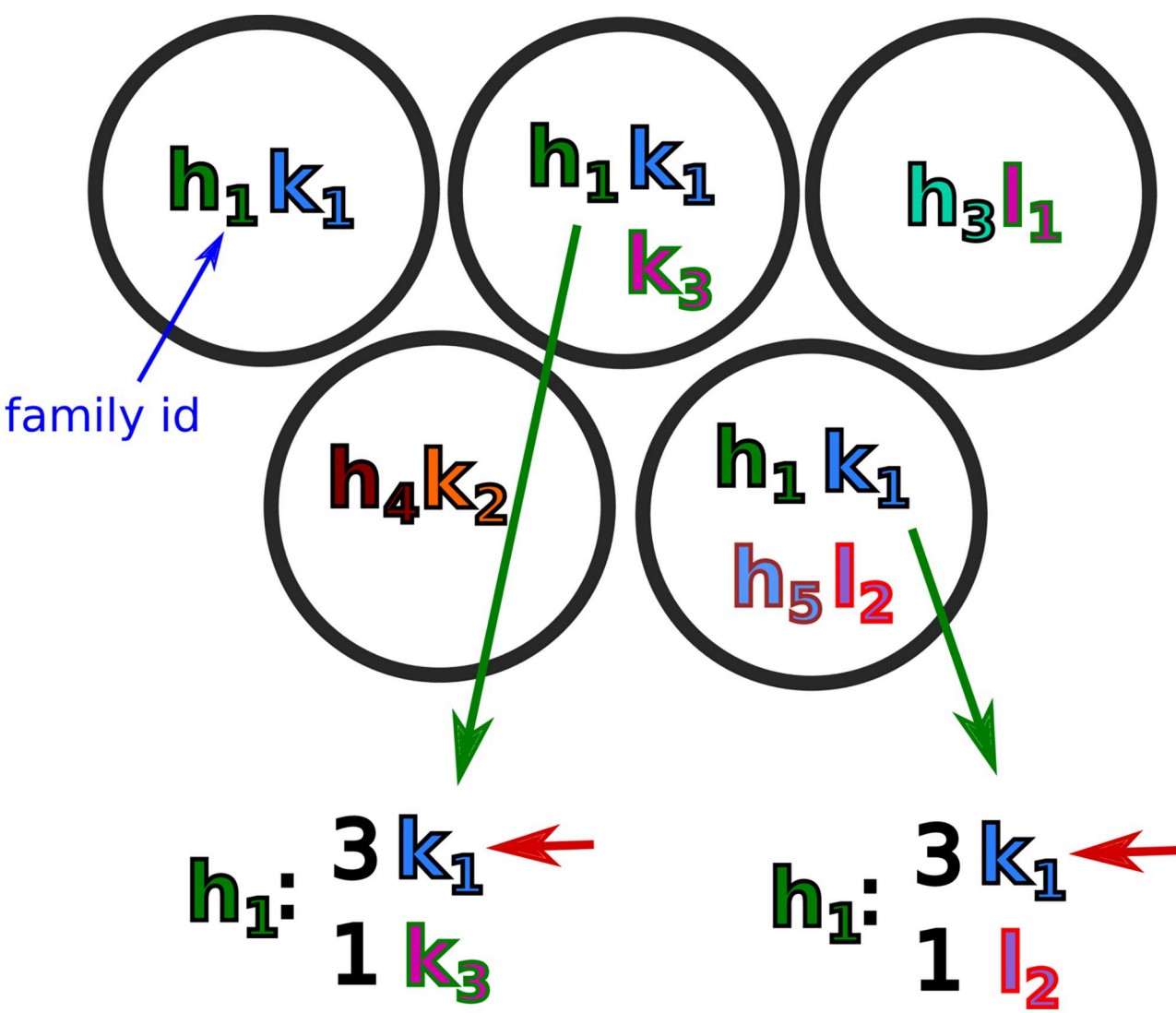

**Fig 11. Schematic representation of the pair info cleaning algorithm showing five droplets, each with some combination of IgH, IgK, and IgL sequences, with subscripts (and color combinations) indicating clonal families (true clonal family id is also indicated as a subscript).** The top left, top right, and bottom left droplets have unique pairings, so require no action. The top middle and bottom right droplets, on the other hand, require disambiguation. In the top middle droplet, $h_1$ has two potential light chain partners: $k_1$ and $k_3$. Since $k_1$ has three $h_1$ family members paired with it across all the droplets, while $k_3$ has only one, we choose $k_1$ (red arrow). A similar procedure is then followed for the bottom right droplet.

cluster $C_h$ we collect the families of all opposite-chain sequences that appear in a droplet with a member of $C_h$, and sort these opposite-chain families by the number of $C_h$ members with which they're paired. For each sequence in $C_h$ with multiple potential partners, we then choose as its final partner the opposite-chain sequence that is a member of the opposite-chain family with the most such "votes". In the case of ties, if the tied sequences are from the same opposite chain family we choose one at random (since they likely stem from the same physical molecule); otherwise we discard all pairing information for the sequence. Whenever we choose a unique partner for a sequence, we remove both sequences from the list of potential partners for all other sequences.

Sequences with zero potential partners can thus arise either in the original data (e.g. from sequencing failure), or as the result of our algorithm's inability to determine a unique partner. In either case, we are left with a sequence that we want to include in the final partition, but for which we have no pair info. For each such sequence we make note of its nearest (by Hamming distance) paired single chain family member, then ignore it during paired clustering. After paired clustering, each such unpaired sequence is then added to the final family of its nearest paired single chain family member. If the unpaired sequence was in a family with no paired sequences, that family remains together; any unpaired singletons remain as singletons. The idea is that nearby sequences, i.e. with shared mutation patterns, come from the same sublineage in the tree. Thus in cases where different sublineages in actuality stem from different, collided rearrangement events, sequences that have no pair info will be able to piggy back on the clustering refinement from those that do.

**Algorithm 1**: Paired clustering method.

```
1 function resolve_clusters(C_l, {C_h^i})
     // Resolve clustering discrepancies between C_l and {C_h^i}:
     reapportion seqs from them into rslvd_clusters, always
     splitting by CDR3, and sometimes also by naive Hamming
     distance.
2   if length of {C_h^i} <2 then
3     return C_l
4   rslvd_clusters: list of resolved clusters to return
5   len_groups: {C_h^i} grouped by CDR3 length
6   for lgrp in len_groups do
7     split_lists: for each cluster in {C_h^i}, a list of clusters with
                   which it should not be merged
8     for c_1, c_2 in all pairs of clusters in lgrp do
9       if naive Hamming distance between c_1 and c_2 > d_0 then
10        Add c_1 to split_lists[c_2]
11        Add c_2 to split_lists[c_1]
12    len_rclusts: resolved clusters for this CDR3 length group
13    for cclust in lgrp do
14      if a cluster c exists in len_rclusts that has no overlap with
         any cluster in split_lists[cclust] then
15        c := c ∪ cclust
16      else
17        Append cclust to len_rclusts
18    Append len_rclusts to rslvd_clusters
19  return rslvd_clusters
20 final_partition: final, joint partition
21 P_h, P_l: single chain heavy and light partitions
22 for C_l in P_l do
23   {C_h^i}: all clusters in P_h that overlap with C_l
24   rslvd_clusters = resolve_clusters(C_l, {C_h^i})
```

```
25  for fclust in final_partition do
26    rf_clusts: clusters in rslvd_clusters that overlap with fclust
27    for rfc in rf_clusts do
28      ovrlp: intersection of rfc and fclust
29      if rfc is the first cluster in rf_clusts then
30        Remove ovrlp from the larger of rfc or fclust
31      else
32        Remove ovrlp from both rfc and fclust
33        Append ovrlp to rslvd_clusters as a new cluster
34    Append rslvd_clusters to final_partition
35  for Cₕ in Pₕ do
36    Same procedure as for Cₗ
```

There are a number of ways to improve on this algorithm. Since our power to determine correct pairings comes from partition information, more accurate partitions would likely improve performance. Because having pair info immediately and dramatically improves partition accuracy, a procedure that iteratively applied pair cleaning and paired clustering would be an easy way to accomplish this. We would first clean pair info using initial single chain partitions, then run paired clustering with this initial pair info, then re-run pair info cleaning with these improved partitions, and so on. An integrated approach, perhaps utilizing a simultaneous likelihood on pair info and partitioning, would probably be even better, although likely also much more difficult to implement. Within each family and droplet in the existing algorithm, it would probably also be better to start with the sequences most likely to be correctly paired, so that once they're successfully paired they can be eliminated from the potential partners of others.

## Approximate bulk pairing

While single cell (paired) data is increasingly easy to produce, bulk (unpaired) data can still be produced with much greater depth. We have thus developed an algorithm that, given single cell and bulk samples drawn from the same pool of B cells, allows to approximately pair sequences from the bulk data using pair info from the single cell data. By "approximately" we mean that it can pair sequences with partners from the correct family, although often not the correct sequence. In many cases, however, this will be sufficient to create a functional antibody.

As with pair info cleaning, we begin with the single chain partitions. For each cluster containing at least one paired sequence, we pair each unpaired sequence with the partner of the nearest (by Hamming distance) paired sequence. Unpaired sequences in clusters with no paired sequences are left unpaired. This has the effect of synchronizing pair info among sequences in sublineages within each cluster, which is important since these single chain clusters will contain significant numbers of collisions.

Because the accuracy of this procedure depends on the accuracy of the partitions, as with pair info cleaning an approach that iteratively applied bulk pairing and paired clustering would likely be significantly more accurate.

## Simulation framework

The implementation of paired heavy/light simulation is based on single chain `partis` simulation (originally described in [46]). For each paired rearrangement event, we simply choose a single tree and pass it to individual heavy and light chain simulation processes. Note that this assumes no biological pairing correlation. We provide here an overview of the single chain simulation process; besides in [46], many more details are available in the manual at https://bit.ly/3GKcjun.

There are two main single chain `partis`-only simulation modes: using a set of parameters inferred from some data sample to mimic that sample as closely as possible; and using a set of reasonable but heuristic priors to simulate *from scratch* without input from any particular sample (together with may ways to combine these). In addition, a naive rearrangement event from plain `partis` can be passed for mutation to the (included) bcr-phylo affinity/fitness-based germinal center simulation package [58]. Unless noted as mimicking a particular data sample, all simulation in this paper is simulated from scratch (see run script for full options https://bit.ly/3Pjll4P).

In all cases, simulation begins with the choice of a set of germline genes from which genes are drawn for each rearrangement in that sample. By default (and for the scratch simulation in this paper) this generates a set of genes and alleles (representing a diploid genome) for each of the V, D, and J regions. The main parameters controlling this generation are, for each locus and region, the number of genes per region and the mean number of alleles for each of these genes (defaults shown in Table 1). Here we define a gene as including all sequences with the same `IMGT` name to the left of the "*", for instance all sequences starting with IGHV1–2 would be alleles of a gene. For each locus and region, we iterate the following procedure to choose alleles for each gene. First we convert the mean alleles per gene (a number $n_a$ between 1 and 2, representing the prevalence of heterozygosity) to a probability of either one ($p_1$) or two ($p_2$) alleles as

$$p_1 = f/(1+f) \tag{1}$$

where

$$f = (n_a - 2)/(1 - n_a) \tag{2}$$

(and $p_2 = 1 - p_1$). After then choosing a number of alleles according to these probabilities, we choose a gene uniformly at random from among all genes with at least this many available alleles. If the chosen gene has more than the necessary number of alleles, we also choose the alleles uniformly at random. Any remaining alleles from the gene are then removed from consideration for future iterations.

**Table 1. Default values for simulation parameters controlling germline set generation when simulating from scratch.**

| parameter | locus | region | value |
|---|---|---|---|
| genes per region | IgH | V | 42 |
| | | D | 18 |
| | | J | 6 |
| | IgK | V | 11 |
| | | J | 3 |
| | IgL | V | 11 |
| | | J | 2 |
| alleles per gene | IgH | V | 1.33 |
| | | D | 1.2 |
| | | J | 1.2 |
| | IgK | V | 1.1 |
| | | J | 1.1 |
| | IgL | V | 1.1 |
| | | J | 1.1 |

After choosing all alleles for all genes, we choose a "prevalence frequency" for each chosen allele, i.e. the fraction of naive rearrangements that will use it. To generate these we start with a minimum desired pairwise prevalence ratio ($r_m$, default 0.1), i.e. the ratio of prevalence frequencies of any two alleles must be greater than this. We then generate an integer "pseudo count" number for each allele by sampling uniformly at random in the interval $[1, 1/r_m]$ and rounding down. The prevalence frequency for each allele is then obtained by normalizing the list of pseudo counts to 1.

Simulation of each rearrangement event then begins by selecting a V, D, and J gene, and deletion and insertion lengths for each gene end or boundary. If mimicking a data sample, these parameter values are all drawn together, according to the probability of that full combination of values in the sample, i.e. taking into account any possible correlation between any of them. When simulating from scratch, on the other hand, values are drawn separately (when correlations are turned off); for example the V 3' deletion length is drawn from a geometric distribution with mean a value typical of data. These are the parameters that define a rearrangement event, and as such they specify a naive sequence.

If mutating with `partis` (rather than passing this naive sequence to bcr-phylo for mutation), we next select a tree. The internal options for generating trees are TreeSim [59] and TreeSimGM [60]; in addition, arbitrary user-generated trees can be passed on the command line. The default TreeSim tree generation (used for scratch simulation in this paper) uses a constant rate birth-death process, conditioned on a fixed age and number of tips, with speciation rate 1 and extinction rate 0.5. The trees generated by all of these methods include mutations along the "root branch" between the naive sequence and the most recent common ancestor of all observed sequences. Mutation itself is handled by the bppseqgen command in Bio++ [61], using a non-default branch ("newlik") that can simulate different rates to different bases (included in `partis` at https://bit.ly/38ItKPj). When mimicking a data sample, the overall mutation rate, as well as the rate to each of the other bases, are specified for each position in each germline gene, with values inferred from each data sample. So, for instance, the $150^{th}$ position in IGHV1−2*02 would use an individual HKY85 model [62] specified by the initial germline base, the overall mutation rate relative to its neighboring positions, and relative rates to each of the four non-germline bases. When mutating from scratch, we use the JC69 model [63] with a four-category Gamma rate distribution, with alpha parameter inferred from [64]. The overall number of mutations for the whole sequence is then drawn from the empirical distribution (unless mutating from scratch, in which case it can be specified in a variety of different ways). Insertion/deletion mutations are added afterwards at the leaves, according to a variety of options specifying their location, frequency, type, and length.

## Performance metrics

The goal in measuring performance of a clustering method on simulation is to quantify the correctness of an inferred partition relative to the known true partition. It is useful to consider separately the two ways that one can miscluster: either by merging a cluster with unrelated sequences, or by splitting apart a true cluster. We can map these two errors onto the usual precision/sensitivity dichotomy by considering correctly (resp. incorrectly) inferred clonal sequences as true (resp. false) positives. While there are several different ways to calculate these numbers [36], we calculate precision as the fraction of sequences in a cluster that are truly clonal (S2 Fig), and sensitivity as the fraction of a true clonal family that end up together, each averaged over all sequences. For brevity, and to emphasize that useful clustering usually requires that precision and sensitivity both be high, we sometimes use their harmonic mean (F1 score).

Validation results on real data differ from simulation in that they perfectly embody the underlying biological processes together with all sources of experimental error; however in real data we also do not in general know the correct clustering. Two approaches have been used to partially avoid this limitation. One approach looks at samples consisting of a single clone, for instance from lymphomas, such that any splitting whatsoever is oversplitting. The other mixes together samples from different subjects, such that any inferred families with sequences from different subjects must be overmerged. The issue with these approaches is that they measure either oversplitting or overmerging alone, whereas a useful clustering method must simultaneously minimize both of these. For instance [19] measures overmerging using inferred families with sequences from multiple donors. Towards the goal of measuring over-splitting, those authors count within-donor "merges," which is a count of all pairs of sequences that appear in the same cluster. As pointed out by those authors, this is not a measure of which of these within-donor merges are correct vs. incorrect, but rather a summary of the distribu-tion of cluster sizes, and in a way that quadratically weights large clusters (S13 Fig). Another issue with the multiple-donor family approach is that it only detects the subset of overmerges between donors, while ignoring overmerges within each donor; because most rearrangement parameters vary between donors (especially germline sets), the observable overmerges are a potentially small subset of all overmerges.

Thus instead of attempting to directly evaluate clustering performance on real data, we instead use it to first evaluate the fidelity of our simulation framework, and second to compare how similarly our method behaves on data and simulation. This approach is especially useful for paired clustering because the heavy and light partitions are, in a sense, independent sources of information on the single true partition. We choose to use cluster size distributions for the second step. While these are poor descriptors of overall clustering quality—they overempha-size the small clusters that dominate real repertoires (but which are often of little experimental interest), and can also obscure large but compensating errors in precision and sensitivity—they visually emphasize the splitting that is at the core of paired clustering.

Disambiguating multiple pairing information (*pair info cleaning*), on the other hand, involves two somewhat separate goals. Correction of the actual pair info is of course important, but the effect of this on clustering performance is also of interest. By construction, our algo-rithm leaves no sequences with multiple partners. After application, then, sequences are either correctly paired, mispaired (with the wrong sequence), or left unpaired. We thus first report these three fractions, then go on to calculate the effect on final clustering performance. Fur-thermore, in some cases a sequence that is not paired correctly, but is paired with a sequence from the correct clonal family, can be useful, since the pairing will often still result in a func-tional antibody. We thus also report the fraction of sequences paired with the correct clonal family, regardless of whether they are paired with the correct sequence in that family. When pairing sequences approximately in bulk samples, we also include sequences that are paired with "similar" families, defined as families with true naive sequences separated by a Hamming distance of three or less. The idea is that sequences from families that are this similar are likely to frequently result in functional antibodies, even if they are from the wrong family.

## Real datasets

We show results on four real data samples from the 10X website [45], with one sample shown in Figs 7 and 8, and S10 Fig, and the other three in https://doi.org/10.5281/zenodo.5860143. We also show pair cleaning results in Fig 8 for a data sample from our collaborators that has not yet been published, but is available at https://doi.org/10.5281/zenodo.5860143.

### General `partis` improvements

While there have been many changes to various parts of `partis` since the original clustering paper, we highlight two here, partly because they have a large impact on speed or accuracy, and partly as representative examples.

**Subcluster annotation.**   As described above and in [46], the `partis` multi-HMM that is used for simultaneous inference on multiple sequences calculates emission probabilities at each position separately for each sequence. This is necessary for computational efficiency: simply multiplying together a number for each sequence is extremely fast. However, it is equivalent to assuming a star tree phylogeny for the family; or equivalently that every mutation in every sequence occurred as a separate mutation event. This is of course a poor assumption for some families, particularly those with highly unbalanced trees (i.e. with large variance in root-to-tip distances).

The most accurate course of action would to incorporate the full phylogenetic likelihood into the emission probabilities. This is what is done by `linearham` [47], which uses a Bayesian phylogenetic HMM to calculate the posterior probabilities of different trees and annotations. While `linearham` is extremely useful for analyzing single lineages, the approach will never be fast enough to run on entire repertoires. We thus developed an iterative *subcluster annotation* scheme that maintains the speed of the star tree assumption, but largely ameliorates the inaccuracy in annotation.

The basic premise of subcluster annotation is that, zoomed in far enough, every tree looks like a star tree. We thus maintain the star tree assumption, but instead of applying it to the entire family of size $N$, we apply it only to subclusters of a certain size $n_s$ by splitting the family into (roughly) $N/n_s$ subclusters using a procedure described below. We then extract the resulting $N/n_s$ inferred naive sequences, and treat them as the observed sequences, again splitting into subclusters of (roughly) size $n_s$. We then iterate until we are left with a single cluster of (roughly) size $n_s$.

Also note that we refer to sizes as "roughly" equal because when dividing a family of arbitrary size into integer-sized subclusters, we cannot, in general, make all subclusters exactly the same size. Note also that the goals of making all subclusters the same size, and of making each of them close to $n_s$, are in conflict. We have chosen to implement it such that the parameter $n_s$ is the maximum cluster size, i.e. if it is set to 3, then all clusters are of size 3 or 2, thereby prioritizing similarity of cluster sizes. For instance, an initial cluster with size 10 would first be split into 4 clusters with sizes 3, 3, 2, and 2; then two with sizes 2 and 2, and finally a single cluster with size 2.

The sequences in each subcluster are determined by the order of sequences in the original cluster; since each cluster is constructed via hierarchical agglomeration while maintaining cluster order during merges, this means that similar sequences tend to end up in the same subcluster. For example, in this size 10 cluster example we would in the first step draw cluster boundaries between the third and fourth sequences, between the sixth and seventh sequences, and between the eighth and ninth sequences.

The end result of this procedure is that the naive sequence inferred on the final cluster is, effectively, the result of weighting by the tree structure, and represents the naive sequence of the entire family.

Subcluster annotation typically takes a similar amount of time to the original star tree method. Annotation time for each [sub]cluster is linear in its size, so each individual subcluster step is parallelized and much faster; but there are now several steps to run, along with some processing infrastructure between steps. For instance annotating a 3000 sequence cluster with the old method takes about 5 seconds; with subcluster annotation this

requires 8 steps, each taking between 0.3 and 1 seconds, resulting in a total time of also around 5 seconds.

**Key translation.**   The multi-HMM, which forces multiple sequences through the same path in the HMM, is an integral part of both the forward and Viterbi calculations used for clustering [46]. The inferred naive sequence from the Viterbi algorithm, for instance, is much more accurate on multiple sequences than on one alone [46]. The accuracy of both the naive sequence (S11 Fig) calculations, however, increase quite quickly with the number of sequences. Because the time required, on the other hand, scales linearly with the number of sequences, we thus implemented a subsampling scheme such that during hierarchical agglomeration we never calculate on more than a relatively modest number of simultaneous sequences. Because this effectively translates the key constructed from the unique IDs for a cluster into a different, smaller, key we refer to it as "key translation".

During hierarchical agglomeration, we thus replace each cluster larger than some size $n_t$ with a smaller cluster consisting of $n_t$ sequences chosen uniformly at random. We found that a value of $n_t$ around 15 was optimal for both the forward calculation and for naive inference (S11 Fig), although both values can be adjusted on the command line if desired. Partly because hierarchical agglomeration involves calculations for a large number of potential cluster merges that do not end up occurring, this technique increases speed by as much as an order of magnitude on samples with large and/or similar clusters (S12 Fig). Upon completion of clustering, for maximum accuracy we calculate the likelihood of and annotation for the final partition using the full clusters (with no key translation).

One complication is that we very commonly encounter a long series of sequential steps that each merge a singleton into a much larger cluster. Each individual step is very unlikely to significantly change either the forward probability or inferred naive sequence, so it is very wasteful to recalculate at every step. For merges in which one cluster is much larger than the other (by default 4 times larger) we thus retain the larger parent's values without recalculation. Keeping track of these translations and propagating them through merging steps is quite involved, but for further details we refer to the relevant section of code https://git.io/JSiOi.

## Supporting information

**S1 Fig. Schematic representation of the space of possible antibodies and antigens, or *rearrangement space*.** The immune system populates the space with naive rearrangements (green crosses) such that any antigen (red squares) will be somewhat near to an antibody. After stimulation by its cognate antigen, a B cell will migrate to a germinal center and undergo affinity maturation to move its offspring closer to the antigen (black tree).
(TIFF)

**S2 Fig. Schematic representation of clustering error in rearrangement space.** If clustering (red circles) is performed on observed (mutated) sequences (black dots), sequences in a family are unnecessarily far from each other. It is much better to cluster on the inferred naive ancestor (blue crosses) of each sequence (and, as clustering progresses, the naive ancestor inferred on the entire cluster at that point in time). Clustering on observed sequences thus conflates inferred SHM (distance from black dots to blue crosses) with inference inaccuracy (distance from blue crosses to green crosses). Note that this figure also shows a potential problem with using shared mutations as a criterion for clustering: while it can help in grouping together members of a sublineage in a high mutation environment, it assumes that all families consist of only a single sublineage, and will thus spuriously split families that do not (such as the five-sequence family at bottom left above). We also show examples of the precision and sensitivity

calculation for one sequence each; these would then be averaged over all sequences to arrive at the values for the entire partition.
(TIFF)

**S3 Fig. Clustering performance on simulation vs. SHM in terms of precision (top) and sensitivity (bottom) for heavy chain (left) and light chain (right).** See Fig 3, which combines these values into the F1 score, for details.
(TIFF)

**S4 Fig. Clustering performance on simulation vs. SHM comparing single-chain to full (paired) `partis` clustering in terms of F1 score (top), precision (middle) and sensitivity (bottom) for heavy chain (left) and light chain (right).** See Fig 3, for details and compare to S5 Fig for SCOPer. Shown also vs. number of families in S7 Fig.
(TIFF)

**S5 Fig. Clustering performance on simulation vs. SHM comparing single-chain to full (paired) `SCOPer` clustering in terms of F1 score (top), precision (middle) and sensitivity (bottom) for heavy chain (left) and light chain (right).** See Fig 3 for details and compare to S4 Fig. for `partis`.
(TIFF)

**S6 Fig. Clustering performance on simulation as a function of the number of families for single chain (no pair info, top) and joint partitions (with paired clustering, bottom) for heavy chain (left) and light chain (right).** In order to focus on the effect of family collisions (unrelated families that are close to indistinguishable), we show performance only in terms of precision, and on samples consisting only of singletons. Each point is the mean (± standard error, often smaller than points) over three samples with 15% mean nucleotide SHM, each consisting of the indicated number of singleton families. See Fig 3 for details, and S7 Fig. to compare single vs. paired `partis`.
(TIFF)

**S7 Fig. Clustering performance on simulation vs. number of families comparing single-chain to full (paired) `partis` clustering in terms of F1 score (top), precision (middle) and sensitivity (bottom) for heavy chain (left) and light chain (right).** See S6 Fig for details, and S4 Fig for comparison vs. SHM.
(TIFF)

**S8 Fig. Effectiveness on simulation of the pair info cleaning method as a function of number of cells per droplet, on samples with a variety of family size distributions in terms of fraction of sequences correctly paired (top) and F1 score of the resulting joint partitions (bottom).** Results shown for samples with family sizes drawn from a distribution inferred from real data (solid red line; corresponds to Fig 4), and where all families have the same, indicated size (dashed lines). Each point is the mean (± standard error, often smaller than points) over three samples, each consisting of 3,000 simulated rearrangement events. With no pair info cleaning, any cells that share droplets (i.e. all points to the right of 1) would have no pair info, which results in performance as shown for single-chain clustering, with very poor IgK precision (S4 Fig middle right, dashed green).
(TIFF)

**S9 Fig. Effectiveness on simulation of the bulk data pairing method as a function of true family size, shown as the fraction of sequences correctly paired (top); and the fraction not correctly paired, split into those mis-paired (bottom left) and left unpaired (bottom right).**

Note that essentially by construction, the fraction correctly paired is simply the paired (non-bulk) fraction of the sample; the goal of the method is to pair sequences with a sequence from the correct (or a similar) family, but not necessarily the correct sequence (Fig 5).
(TIFF)

**S10 Fig. Comparison of inferred parameter distributions on a real data sample (green) and the corresponding true distributions in a simulation sample generated using parameters inferred from that real data sample (red) (see details in Fig 7).** Shown here are cluster size distributions (top left), D gene usage (top middle), D 5' deletion lengths for all D genes together (top right), per-position SHM frequencies for IGHJ3*02 (bottom left), number of J segment mutations (over all J genes, bottom middle), and sequence amino acid content (bottom right). Distributions for all other parameters, and for the same studies performed on three other real data samples, may be found at https://doi.org/10.5281/zenodo.5860143.
(TIFF)

**S11 Fig. Naive sequence inference accuracy on simulation as a function of family size.** Accuracy is measured as the Hamming distance separating the true and inferred naive sequences. Each point is the mean (± standard error) over three samples, each consisting of 50 simulated rearrangement events with the indicated size.
(TIFF)

**S12 Fig. Time required for clustering with a single process vs. the maximum calculated cluster size on samples with 5000 sequences divided among the indicated number of families, each restricted to the same gene combination and CDR3 length to eliminate irrelevant families.** Clusters larger than (approximately) the indicated size are subsampled for the Viterbi and forward calculations during clustering (see text).
(TIFF)

**S13 Fig. Cluster size distributions for `partis` and `enclone` on real data from [44] with (left) and without (right) log y axis.** While the overall distributions are similar, `enclone`'s largest clusters are significantly larger. Because the "within-donor merges" column effectively squares the cluster size, these largest clusters dominate the in Table 1 of [19], which is why this number is much larger for `enclone` than `partis` (2.2 vs 1.6 million), despite `partis` having larger clusters for much of the distribution (as can be seen in the linear y plot, `enclone` has $\simeq$2% more singletons, which is why with log y the `partis` line can be seen to be higher for most of the middle of the distribution). Most large `enclone` clusters can be constructed by merging several smaller `partis` clusters, then splitting off some fraction of singletons (https://doi.org/10.5281/zenodo.5860143); these two dynamics explain why `enclone` has many more within-donor merges (which depend almost entirely on the largest few clusters), and likely also why `enclone` has relatively poor sensitivity in our simulation tests (since many singletons are split from their correct family).
(TIFF)

## Acknowledgments

We would first like to thank our collaborators who created data that was used during development of these methods: Leslie Goo, Lisa Levoir, Jay Lubow, Julie Overbaugh, Jamie Guenthoer, Michelle Lilly, and Zak Yaffe. We would also like to thank the Kleinstein lab for help with `SCOPer`, Wyatt McDonnell and David Jaffe for discussions and help with `enclone`, and Kristian Davidsen for help editing the manuscript, as well as for suggesting the possibility of approximately pairing bulk data with a matched single cell sample.

## Author Contributions

**Conceptualization:** Duncan K. Ralph, Frederick A. Matsen, IV.

**Data curation:** Duncan K. Ralph.

**Formal analysis:** Duncan K. Ralph.

**Funding acquisition:** Frederick A. Matsen, IV.

**Investigation:** Duncan K. Ralph.

**Methodology:** Duncan K. Ralph.

**Project administration:** Frederick A. Matsen, IV.

**Resources:** Frederick A. Matsen, IV.

**Software:** Duncan K. Ralph.

**Supervision:** Frederick A. Matsen, IV.

**Validation:** Duncan K. Ralph.

**Visualization:** Duncan K. Ralph.

**Writing – original draft:** Duncan K. Ralph, Frederick A. Matsen, IV.

**Writing – review & editing:** Duncan K. Ralph, Frederick A. Matsen, IV.

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
