## [Decision Letter · Decision Letter 0]

17 May 2022

Dear Dr Ralph,

Thank you very much for submitting your manuscript "Inference of B cell clonal families using heavy/light chain pairing information" for consideration at PLOS Computational Biology.

As with all papers reviewed by the journal, your manuscript was reviewed by members of the editorial board and by several independent reviewers. In light of the reviews (below this email), we would like to invite the resubmission of a significantly-revised version that takes into account the reviewers' comments.

As you can see from the review reports, all referees agreed that you present a very valuable contribution to the field of B cell clonal inference. Referee #3 raised a few important points that I would very much encourage you to address. In particular, they raised issues relating to the incompleteness discussion of previous related methodological work, the way the method has been benchmarked, and the simulated data that have been generated for benchmarking. Additionally, they remarked that data could be shared together with a more detailed description of how they were generated, either in the form of code or a more comprehensive describing text.

We cannot make any decision about publication until we have seen the revised manuscript and your response to the reviewers' comments. Your revised manuscript is also likely to be sent to reviewers for further evaluation.

Sincerely,

Roland R Regoes

Associate Editor

PLOS Computational Biology

Thomas Leitner

Deputy Editor

PLOS Computational Biology

Reviewer's Responses to Questions

**Comments to the Authors:**

Reviewer #1: In this paper, Ralph and Matsen study the very important question of Inference of B cell clonal families using heavy/light chain pairing information. Given the increase of available single-cell data, clonal family inference based on paired chain data is an important topic that has remained underrepresented so far in the literature. The authors present a very detailed study of this question. My questions mostly pertain to the clarity of the text.

- the text is extremely detailed, which is great. However, given its extensive nature, it’s at times hard to read. Can you try to shorten and make it more concise (intro and results mostly).

- In order to be able to perform paired-chain clonal family clustering, a few preprocessing steps are required as outlined by you. Can you make a general diagram wherein you detail all the steps that contribute to paired chain clonal inference (e.g., including data cleaning, and how bulk-single cell matching can contribute to VH-VL clonal family constructions etc)?

- I did not find a mention of how the simulated data was generated. If it’s not in the text yet, please add a description in the methods section.

- lines 179 – 189 don’t seem to really reflect the title. Can you align these?

- can you better define what you mean by partition? Also, you use the word partition a lot…so it should also be in figure 1?

- Can you make a figure for the performance metrics (purity etc)..will make it easier to understand (probably just expand Figure S2 a bit?)?

- For the Bulk and single cell data augmentation investigation —> which data was used for that?

- Can you add a datasets section to the methods section if not done already wherein you describe the experimental data used.

- Do the conclusions you get on single cell data hold across different single-cell datasets? Are they dependent on B cell type (memory, plasma, naive)?

- You show that facility size is an important factor for obtaining correct pairing information. What’s the distribution of family sizes within a given repertoire (how stable is this distribution across samples) —> based on this distribution and the accuracy values per family size, which what confidence?/accuracy can we reconstruct pairing in a given repertoire (in figure S9 you show that family size in repertoires is mostly low…which means accuracy of VH-VL reconstruction will also be low?)?

Reviewer #2: This manuscript by Ralph and Matsen reports a new approach for combining information from immunoglobulin heavy and light chains to refine the accuracy of B cell clonal assignments. As clonal assignment is a critical prerequisite for many downstream analyses, improvements are always a welcome advance. Moreover, the technique is timely, as well, with the rapidly increasing adoption of single-cell sequencing techniques. In addition, Ralph and Matsen describe potentially impactful algorithms for “pair cleaning” of droplets containing multiple cells and combining single-cell (paired) with bulk (unpaired) sequencing data. I suspect that in the long run, these may prove even more useful than the refinement to clonal assignment. Overall, this manuscript represents a solid piece of work that will help the field continue to advance.

Minor comments:

1. Line 63: LAIR1 insertion is thought to be an SHM-mediated event, not a part of V(D)J recombination. Evidence in the cited papers includes LAIR1 insertions in the switch region that are spliced into the mature mRNA between the variable and constant regions.

2. Line 102: Is this the intended reference here? Given that the sentence is about the number of cells, rather than the diversity of receptors, better references might be https://doi.org/10.1111/j.1365-2249.2010.04206.x and https://doi.org/10.1002/cyto.b.20547.

3. Line 111: “0.9” needs to be labeled as referring to sequence identity (as in lines 260-261).

4. Line 127: I would add something like “have been _typically_ sequenced,” to account for occasional methodological contortions as in refs 8 and 19.

5. Lines 146-149: “mature collisions” are not clearly illustrated in S2 Figure; I assume the intention is the two middle red circles, but this could be highlighted better.

6. Fig. 1: It’s a little confusing that only the two a’s are labeled “droplet id”. Perhaps this annotation could be left only to the figure caption, or at any rate pointed toward discordant droplet ids.

7. Line 289: I believe this should refer to S2 Figure (instead of S1).

8. Lines 300-302: This sentence should be rewritten (and perhaps expanded slightly) for clarity.

9. Lines 308-309: 10x doublet frequencies depend intimately on loading levels; 0.8% is the rate when 1,000 cells are loaded. At more typical loading levels, the company estimates 3-6% multiplets. See eg https://kb.10xgenomics.com/hc/en-us/articles/360059124751-Why-is-the-multiplet-rate-different-for-the-Next-GEM-Single-Cell-3-LT-v3-1-assay-compared-to-other-single-cell-applications-; even though that’s 3’ instead of 5’, I think they are probably similar. Of course, not all of these will appear in the sequencing data as having ambiguous pairing, but I still think it’s important set the scope properly.

10. Fig 3: Why are these simulations done with 3,000 rearrangement events instead of 10,000? Also, is 12,000 the total number of heavy+light chains, or is it 12,000 each?

11. Lines 319-322: What happens if the number of cells per droplet is modeled (more accurately) as a Poisson distribution, instead of a uniform number? I assume this will actually result in better performance, but the authors may have better intuition here.

12. Lines 324ff: Many single-cell antibody experiments pre-enrich the input B cells for reactivity with an antigen of interest, which will presumably have a large effect on the family size distribution. I doubt this will substantially impact the results here, but worthwhile for the authors to note.

13. Lines 328-330: The description of S7 Figure here is impeded by the unfortunate placement of Fig 4 immediately after – it took me a couple of reads to realize why the text and image didn’t match.

14. Fig 4: Is the right panel calculated mutually between heavy and light, or separately and then added together? I ask because a scenario in which the similar-family rate is significantly higher for heavy chains that light chains seems plausible to me and should be explored/addressed by the authors. This might be especially important for comparing results with ref 32. A similar question might also apply to the left panel, as well as the top right panel of Fig. 3, though I am less concerned in those cases.

15. Lines 350-354: I was still thinking of S8 Figure when I read these lines and was confused by the apparent contradiction between the text and the bottom left panel. Perhaps another panel could be added to S8 that explicitly shows what is being described here, so as to prevent confusion.

16. Fig 5: Should the bottom panel include a line for paired SCOPer?

17. Lines 369-370 (and elsewhere): Please don’t forget to update this DOI.

18. Lines 443-446: Something is wrong here, though I’m not sure exactly what. Perhaps “inconsistent” at the end of line 443 should be “consistent”?

19. Lines 445-446: Isn’t the more relevant comparison between the solid red line in the top left vs the one in the bottom left? Either way, I’d like to see an additional supplemental figure parallel to S4, but with family size on the x-axis instead of SHM.

20. Line 636: I don’t think I am understanding what it means to “remove both sequences.” In Fig. 10, both droplets are shown as having 3 “k1” pairings, suggesting it hasn’t been removed after the first one was disambiguated.

21. References 16, 21, 33, and 40: Journal name is corrupted/missing.

22. S8 Figure: there seems to be an issue with the way lines at zero are displayed in the bottom two panels.

Reviewer #3: Ralph & Matsen describe an extension of their previous work on single immunoglobulin chain B cell clonal inference into the setting of paired chains, and benchmark their method against some existing ones. They also describe an algorithm that attempts to tease out the "true" immunoglobulin pair in a cell that happens to contain more chains than expected. In addition, they describe a procedure that matches an unpaired chain approximately to a paired BCR. Overall, paired chain clonal inference is a welcoming addition to the field, especially in light of the ever increasing volume of single-cell paired BCR data, along with continued efforts of bulk BCR-sequencing. Several issues of concern are noted below, mainly the omissions of relevant developments, the inclusions of which for comparison would be highly informative. On the other hand, I am of the opinion that there is very little novelty about the "approximate bulk pairing" procedure.

The authors do a commendable job in the Introduction covering a wide range of immunological backgrounds relevant to the task of clonal inference, albeit with several scientifically important omissions and inaccuracies, some of which have implications for their subsequent benchmarking work and beyond.

First, the authors claim that the widespread practice when it comes to clonal clustering is to do so based on CDR3 sequence with a threshold that is "typically 0.9 [11]" (L111). Aside from the fact that there is no mentioning whether this is a threshold for nucleotide sequences or amino acid sequences, it is highly questionable whether 0.9 is a "typical" threshold. The reference given, [11], is not a review article; nor is it an acceptable reference to support this claim. I could not find any discussion regarding clustering threshold in [11]. If anything, [11] speaks against 0.9 being typical as [11] appears to be using a threshold of 1 -- "Clonotypes were identified on the basis of ... identical amino acid sequences of the heavy-chain CDR3". It has been my observation that many works in the field tend to use a threshold -- nucleotide or aa -- that is lower than 0.9. Examples abound, but just to list a few: Galson et al. use 0.8 (nucleotide); Vander Heiden et al. use 0.75 (nucleotide); Nielsen et al. use 0.85 (aa). On the other hand, Soto et al. and Briney et al. both use 1 (aa). In addition, by claiming that 0.9 is "typical" and proceeding to use for some of the benchmarking such a high threshold that many in the field do not use, the authors risk artificially deflating the sensitivity performance of the "VJ CDR3" method.

Second, the coverage on current clonal inference methods is far from being complete. Given that the authors note in Discussion that their paired clustering could likely improve if they "used pairing information from the start" (L457), an important fact missing from their survey is that there have already been works that use paired heavy and light chains "from the start" for clonal inference. Some examples include Soto et al., Alsoussi et al., and Kim et al., all of which use heavy and light chains simultaneously when partitioning paired BCRs -- as opposed to partitioning by heavy first, then by light chains, which is what the SCOPer method benchmarked does. One-step partitioning could be of interest for benchmarking as well, in addition to the benchmarked two-step partitioning.

Given that the authors state that a fundamental issue affecting existing methods is that somatic hypermutation tends to cause biological clones to be computationally split (L115), another notable omission from Introduction is therefore the development of methods that attempt to address exactly this issue. Nouri & Kleinstein (2020), for instance, developed an approach that makes use of shared SHMs when inferring clonal relationships. This method is not cited, but should be. In fact, unless mistaken, this method is accessible via SCOPer through specifying `vj="vj"` in the very script (https://bit.ly/3K8KfBa) that the authors use on L268 (In the link, `vj` appears to have been set to "novj", meaning no SHM information is to be used). `enclone` also attempts to address the SHM issue (see below).

More importantly, given that the authors claim that their previously published work on single-chain clonal inference [12], to which the current manuscript appears to be a paired-chain extension, addresses the abovementioned SHM issue (L120), it seems to me that the current setup of their benchmarking -- partis vs. a variant of SCOPer (spectral clustering, single chain or paired) that doesn't use SHM information -- is unfair, as it compares a method designed to overcome the SHM issue, against a method for which the SHM issue is a known problem. A fairer comparison, in my opinion, would thus be to compare partis against the variant of spectral clustering that does take into account of shared SHMs, i.e. the method proposed by Nouri & Kleinstein (2020) and implemented in SCOPer.

A related note regarding benchmarking is the seeming discrepancy in the performance of some of the methods benchmarked here, compared to elsewhere. For instance, one of the methods that the authors benchmark partis against, spectral clustering without using SHM information [17], is reported to have median sensitivity in the range of 96%-98% when benchmarked in the original publication [Fig 9 of ref. 17; specificity was reported instead of precision] against repertoire data simulated by Ralph & Matsen, 2016 [12] with 0.10 fraction of SHM. In Fig S5, however, the authors report that single chain scoper [17] has "igh: sensitivity" around 0.9 at SHM level of 0.10. The magnitude of performance discrepancy of the same method in two separate benchmarking settings brings the question as to how much of the better performance by partis and worse performance by [17] can be attributed to the inherent differences in the simulated repertoire data used for benchmarking. In other words, is the seemingly improved performance of partis reported here going to hold when applied on the simulated data used in [17], or, as suggested earlier in the review, the mutated repertoire data simulated using AbSim used in Fig 5 of Nouri & Kleinstein (2020) if the benchmarking were to be performed against spectral clustering using shared SHM information? After all, there is little to no description in the current manuscript as to how the authors generated their simulated datasets this time. It thus seems not only fair but necessary to perform benchmarking on not only the authors' own simulated data, but also the alternative methods' original simulated data.

Another major omission from existing clonal clustering methods that would be particularly relevant for benchmarking is `enclone` from 10x Genomics (https://support.10xgenomics.com/single-cell-vdj/software/pipelines/latest/algorithms/clonotyping/), especially given the current widespread adoption of 10x Genomics' products for single-cell paired BCR profiling, as well as the fact that `enclone` is built into `cellranger`, meaning that anyone processing 10x Genomics data with `cellranger` would have access to clonal inference results produced by `enclone` (whether they use those results is a separate question). Granted, a formal `enclone` preprint was only recently posted by Jaffe et al.; however, `enclone` itself has been in production since cellranger 5.0.0 as of 2020-11-19. Of note, `enclone` utilizes both heavy and light chains, as well as shared SHM when inferring clones, making it a particularly useful candidate for benchmarking. In addition, while not exactly the same as the "pair info cleaning" that authors propose here, `enclone` also handles the presence of multiple immunoglobulin chains.

On disambiguating multiple pairing information, the authors claim that cells with more-than-expected number of chains are "typically discarded [36]" (L311). Several recent studies, in contrast, strike a middle ground between straight out discarding and preserving everything. For instance, Jiang et al. perform "pair info cleaning" before clonal inference by preserving the most abundantly sequenced heavy chain as indicated by UMI count. Lee et al., taking a different approach, keep the lambda chain and discard the kappa chain when both are present, reasoning that lambda rearrangement only happens if kappa failed. Both approaches are not without biological basis, and should be referenced. Benchmarking comparison would be of interest if at all possible. In particular, in cases where both kappa and lambda chains are present in a cell, should the vote-by-majority approach suggested by the authors (L632) be in favour of keeping the kappa chain, it seems not unreasonable to say that Lee et al.'s approach would produce a result more in line with the actual underlying B cell development.

I find the descriptions of the results of validating simulation against real data to involve too much hand waving. The authors describe the clone size distributions from simulated data and real data (Fig 6) as sufficiently similar to provide confident validation (L373-374). Looking at Fig 6, the actual level of similarity seems to be a rather subjective matter, especially when compared to S9 Figure. In the latter, there is little doubt that the curves for real and simulated data match ever so closely. In Fig 6, on the other hand, the differences are quite pronounced. It would be informative to quantify the comparison of distributions using a metric. More importantly, to convincingly demonstrate that the simulation, with the parameters used, reasonably and sufficiently matches real data, it would be important to have some form of "negative control" -- for example, simulated data generated with parameters that are wildly different from what's expected of reality.

In general, there is very little detail on how simulations were performed, both for paired chain clonal inference, and for pair info cleaning. This makes it hard to evaluate whether the simulated datasets are appropriate, and impossible for future readers to replicate such simulations. The Zenodo link provided that is supposed to contain simulated data appears to be a simple data dump with no metadata description whatsoever, and as such is not at all helpful in this regard.

The authors try to match bulk-sequenced unpaired chain with single-cell paired chains using clonal family information. They call this "approximate bulk pairing", emphasizing that the unpaired chain can be matched with an opposite chain from the correct clonal family, but not necessarily the correct sequence. They argue that this is in practice "sufficient to create a functional antibody" (L664-671). In essence, their algorithm matches an unpaired chain with the pair of chains containing the unpaired chain's nearest neighbour in the sequence space. While the preparatory work leading up to this matching step -- that is, single chain partitioning and paired chain partitioning described in earlier sections -- is novel, this matching algorithm itself is not novel (L454). The practice of using single-cell paired chains as anchor to "fish" amongst bulk unpaired chains has been used both in academia and in industry for antibody discovery. Multiple COVID-19 works use this strategy. Kim et al., for instance, paired bulk-sequenced bone marrow heavy chains with light chains from single-cell plasmablasts with paired chains from the same single chain partition, expressed the antibodies, and tested for neutralization.

--- Minor ---

L180-181: "as well as a paired method that has been used [34]" While Hoehn et al. [34] did use said method in their 2021 work, the same method was first used in 2019 by Oh et al., which would be the correct reference instead. From Oh et al.: "In addition, Igh V(D)J sequences were further grouped based on whether the associated cells shared common combinations of Igk V with Igk J or Igl V with Igl J gene annotations." This step was done via the `light_cluster.py` script from Immcantation, which [34] also used.

Fig 2 caption: "All methods are publicly available except for SCOPer, whose paired method was obtained via personal communication." It is factually incorrect to say that the SCOPer paired method is not publicly available. The script "obtained via personal communication", referenced on L268 at https://bit.ly/3K8KfBa, is nothing more than a wrapper making calls to functions all of which are available on CRAN. In fact, to use the paired method, one need not use the script at all. As long as one specifies `only_heavy=FALSE` to either `hierarchicalClones` or `spectralClones`, both functions from the publicly available SCOPer R package, one could use the paired method. Just like a user would have to specify different parameters to use partis, the program written by the authors, to perform a task -- such as clonal clustering using paired information as covered in this manuscript, having to specify parameters to use SCOPer does not make it publicly unavailable.

Fig 4, left vs. S8 Figure, top: These appear to show results for similar, if not identical, benchmarking comparisons. However, the patterns and numerical range are completely different for all the curves except for the one corresponding bulk data frac = 0. What is going on?

L296-298 "a lot", "a little" -- More rigourous and quantitative descriptions should be used instead of casual colloquialism.

L300-302: "we show performance as a function of the distance between naive rearrangements only versus the number of rearrangement events (S6 Figure)." I could not find a S6 Figure matching the text description on L300-302. The S6 Figure present shows "Clustering performance on simulation as a function of the number of families". Could there have been a mix-up?

L215 "they perfectly recapitulate the underlying biological processed" -- Is this meant to be "imperfectly"?

L470-472: "We also note that we have used the pair cleaning algorithm with good results on (not-yet-published) non-10X data that flow-sorted cells into wells, but left multiple cells in some wells." It is of course up to the editor to decide, but I do not find an unsubstantiated statement such as this one that is essentially "data not shown" based on "not-yet-published" data to be acceptable for peer-reviewed publication.

Last but not least, I find the entire section of "General partis improvements", including Fig 8, to be completely out of place. Whilst partis might be an all-encompassing software that performs V(D)J annotation, clonal inference, and other tasks, I do not find it relevant or appropriate to piggyback updates concerning V(D)J annotation, a task upstream of clonal inference, into a manuscript that is supposed to be about an extension of single chain clonal inference into the paired chains setting. I suggest that this section be taken out and submitted elsewhere, for instance, perhaps as an Application Note at Bioinformatics. Again, this is, of course, up to the editor.

--- References ---

Galson et al. Deep Sequencing of B Cell Receptor Repertoires From COVID-19 Patients Reveals Strong Convergent Immune Signatures. Front. Immunol., 2020.

Vander Heiden et al. Dysregulation of B Cell Repertoire Formation in Myasthenia Gravis Patients Revealed through Deep Sequencing. J. Immunol., 2017.

Nielsen et al. Human B Cell Clonal Expansion and Convergent Antibody Responses to SARS-CoV-2. Cell Host & Microbe, 2020.

Soto et al. High frequency of shared clonotypes in human B cell receptor repertoires. Nature, 2019.

Briney et al. Commonality despite exceptional diversity in the baseline human antibody repertoire. Nature, 2019.

Alsoussi et al. A Potently Neutralizing Antibody Protects Mice against SARS-CoV-2 Infection. J. Immunol., 2020.

Kim et al. Germinal centre-driven maturation of B cell response to mRNA vaccination. Nature, 2022.

Nouri & Kleinstein. Somatic hypermutation analysis for improved identification of B cell clonal families from next-generation sequencing data. PLOS Comp. Bio., 2020.

Jaffe et al. enclone: precision clonotyping and analysis of immune receptors. BioRxiv, 2022. https://www.biorxiv.org/content/10.1101/2022.04.21.489084v1

Jiang et al. Single-cell repertoire tracing identifies rituximab-resistant B cells during myasthenia gravis relapses. JCI Insight, 2020.

Lee et al. Long-lasting germinal center responses to a priming immunization with continuous proliferation and somatic mutation. BioRxiv, 2021. https://doi.org/10.1101/2021.12.20.473537

Oh et al. Migrant memory B cells secrete luminal antibody in the vagina. Nature, 2019.

**Have the authors made all data and (if applicable) computational code underlying the findings in their manuscript fully available?**

Reviewer #1: Yes

Reviewer #2: Yes

Reviewer #3: **No: **Simulated data is provided but without any associated metadata that makes it usable. No code and little-to-no method description in text are provided for generating the simulated data.

PLOS authors have the option to publish the peer review history of their article (what does this mean?). If published, this will include your full peer review and any attached files.

Reviewer #1: No

Reviewer #2: **Yes: **Chaim A Schramm

Reviewer #3: No
---

## [Decision Letter · Decision Letter 1]

28 Sep 2022

Dear Dr Ralph,

Thank you very much for submitting your manuscript "Inference of B cell clonal families using heavy/light chain pairing information" for consideration at PLOS Computational Biology. As with all papers reviewed by the journal, your manuscript was reviewed by members of the editorial board and by several independent reviewers. The reviewers appreciated the attention to an important topic. Based on the reviews, we are likely to accept this manuscript for publication, providing that you modify the manuscript according to the review recommendations.

As you will see two of the three reviewers are happy with your revisions. Reviewer #3 raises a couple of points relating to how previous work is being discussed. The first point concerns the level of validation. In the view of the editor a biological validation is beyond the scope of the study, but the reviewers criticism could be adequately addressed by calling the validation "computational validation", "validation by simulation" or similar. We also encourage you to consider providing a description of the clustering in the study by Soto et al. more nuanced than captured by the phrase "by hand".

Sincerely,

Roland R Regoes

Academic Editor

PLOS Computational Biology

Thomas Leitner

Section Editor

PLOS Computational Biology

Reviewer's Responses to Questions

**Comments to the Authors:**

Reviewer #1: The authors have addressed all my comments.

Reviewer #2: The authors have done a thorough job of responding to the critiques and suggestions of all three reviewers. I have no further comments at this time.

Reviewer #3: The revised submission constitutes an improved manuscript. The authors attempted to address most comments reasonably and sufficiently. Two minor points remain objectionable.

The authors tried to frame as an advantage of their approximate bulk pairing method that it "provides a codified, repeatable method together with validation" [L421, version showing changes]. Soon after, nonetheless, they note that "[o]ur approximate bulk pairing method would also benefit from evaluation on real data samples; here we have only tested it on simulation" [L455-456]. First, the so-called "validation" for the approximate bulk pairing method was purely computational, based on simulation, and using one of various metrics possible for evaluation -- in this case, similarities in clone size distribution (Fig 4). Whether that is the optimal metric for benchmarking is by no means a given and a separate question. In contrast, the existing work cited, [52], carried out actual, real-world, biological validation. Antibodies that were formed by computationally pairing heavy and light chains were experimentally synthesized, functionally characterized, and confirmed to bind and neutralize SARS-CoV-2 spike. When it comes to validation of BCRs/antibodies, the only gold standards are functional assays. Otherwise, there is simply no telling whether the pairing suggested by this approximate bulk pairing method would give rise to an experimentally viable antibody that can be successfully expressed, remain stable, and bind its cognate antigen. After all, if not for binding, what's the point of having a paired BCR at all? In other words, there was no biologically meaningful "validation" here [L421] and the method indeed could benefit from actual experimental validation that is more robust than what's proposed on L455-456.

In comparing with other prior works [50-52], the authors also casually suggested that these works clustered paired data "by hand" [L381]. It is understandable that when trying to promote one's own method, one would like to highlight the perk of their method over others'. But this is just silly and grossly inaccurate. [50] reported 9-17 million B cell clonotypes; I find it extremely unlikely that Soto et al. sat down and painstakingly went through millions of BCRs, sequence by sequence, "by hand", to cluster them. Clearly Soto et al. would have used some program or implemented some custom code, both scenarios already covered by "with non-public code" [381]. In short, I suggest that the inaccurate characterization of others' works, "by hand", be dropped.

**Have the authors made all data and (if applicable) computational code underlying the findings in their manuscript fully available?**

Reviewer #1: Yes

Reviewer #2: Yes

Reviewer #3: Yes

PLOS authors have the option to publish the peer review history of their article (what does this mean?). If published, this will include your full peer review and any attached files.

Reviewer #1: No

Reviewer #2: **Yes: **Chaim A Schramm

Reviewer #3: No

Figure Files:

Data Requirements:

Reproducibility:

References:

---

## [Editor Report · Decision Letter 2]

9 Nov 2022

Dear Dr Ralph,

We are pleased to inform you that your manuscript 'Inference of B cell clonal families using heavy/light chain pairing information' has been provisionally accepted for publication in PLOS Computational Biology.

Best regards,

Roland R Regoes

Academic Editor

PLOS Computational Biology

Thomas Leitner

Section Editor

PLOS Computational Biology

---

## [Editor Report · Acceptance letter]

21 Nov 2022

PCOMPBIOL-D-22-00446R2 

Inference of B cell clonal families using heavy/light chain pairing information

Dear Dr Ralph,

I am pleased to inform you that your manuscript has been formally accepted for publication in PLOS Computational Biology. Your manuscript is now with our production department and you will be notified of the publication date in due course.

With kind regards,

Zsofia Freund
